# Immune suppressive landscape in the human esophageal squamous cell carcinoma microenvironment

Yingxia Zheng [1,2,10 ✉], Zheyi Chen[1,10], Yichao Han [3,10], Li Han[1,10], Xin Zou[4], Bingqian Zhou[1], Rui Hu[5], Jie Hao[4], Shihao Bai[4], Haibo Xiao[5], Wei Vivian Li[6], Alex Bueker[7], Yanhui Ma[1], Guohua Xie[1], Junyao Yang[1], Shiyu Chen[1], Hecheng Li [3 ✉], Jian Cao [7,8 ✉] & Lisong Shen [1,9 ✉]

Cancer immunotherapy has revolutionized cancer treatment, and it relies heavily on the comprehensive understanding of the immune landscape of the tumor microenvironment (TME). Here, we obtain a detailed immune cell atlas of esophageal squamous cell carcinoma (ESCC) at single-cell resolution. Exhausted T and NK cells, regulatory T cells (Tregs), alternatively activated macrophages and tolerogenic dendritic cells are dominant in the TME. Transcriptional profiling coupled with T cell receptor (TCR) sequencing reveal lineage connections in T cell populations. CD8 T cells show continuous progression from pre-exhausted to exhausted T cells. While exhausted CD4, CD8 T and NK cells are major proliferative cell components in the TME, the crosstalk between macrophages and Tregs contributes to potential immunosuppression in the TME. Our results indicate several immunosuppressive mechanisms that may be simultaneously responsible for the failure of immuno-surveillance. Specific targeting of these immunosuppressive pathways may reactivate anti-tumor immune responses in ESCC.

[1] Department of Laboratory Medicine, Xin Hua Hospital, Shanghai Jiao Tong University School of Medicine, Shanghai, China. [2] Institute of Biliary Tract Diseases Research, Shanghai Jiao Tong University School of Medicine, Shanghai, China. [3] Department of Thoracic Surgery, Ruijin Hospital, Shanghai Jiao Tong University School of Medicine, Shanghai, China. [4] Key Laboratory of Systems Biomedicine (Ministry of Education), Shanghai Centre for Systems Biomedicine, Shanghai Jiao Tong University, Shanghai, China. [5] Department of Thoracic Surgery, Xin Hua Hospital, Shanghai Jiao Tong University School of Medicine, Shanghai, China. [6] Department of Biostatistics and Epidemiology, Rutgers School of Public Health, New Brunswick, NJ, USA. [7] Rutgers Cancer Institute of New Jersey, New Brunswick, NJ, USA. [8] Department of Medicine, Robert Wood Johnson Medical School, Rutgers University, New Brunswick, NJ, USA. [9] Faculty of Medical Laboratory Science, Shanghai Jiao Tong University School of Medicine, Shanghai, China. [10] These authors contributed equally: Yingxia Zheng, Zheyi Chen, Yichao Han, Li Han. ✉email: zhengyingxia@xinhuamed.com.cn; lihecheng2000@hotmail.com; jian.cao@cinj.rutgers.edu; lisongshen@hotmail.com

Esophageal cancer is one of the most commonly diagnosed and deadly cancer types, especially in East Asia[1]. However, esophageal cancer is significantly understudied compared with other common tumor types, and in the recent decades there has been limited progress in therapeutics. Histologically, esophageal cancer can be classified into two subtypes: adenocarcinoma (EAC) and squamous cell carcinoma (ESCC)[2]. ESCC is the dominant subtype and accounts for ~90% of esophageal cancer cases worldwide[3]. Esophageal cancer is also among the tumor types with the highest median mutation burden, and is ranked higher than kidney, head and neck, and colorectal cancer[4]. Recently, PD-1 antibodies, pembrolizumab and nivolumab have been used in clinical trials for subsets of patients with advanced ESCC for whom first-line chemotherapy failed. However, they show only moderate improvement in the overall survival compared with chemotherapy[5,6]. A systematic interrogation of infiltrating immune cells in ESCC will help to profile the immune status of ESCC, evaluate the application of current checkpoint blockades, and, most importantly, lead to innovative immunotherapies.

Single-cell transcriptome analysis of immune cells in tumors provides a way to comprehensively study these cells in a highly complex tumor microenvironment (TME). Recently, single-cell RNA sequencing (scRNA-seq) has been applied to tumor-infiltrating immune cells isolated from limited types of cancers, including cutaneous melanoma[7,8], non-small cell lung cancer[9,10], hepatocellular carcinoma[11,12], basal cell carcinoma[13], colorectal cancer[14,15], and breast cancer[16]. These studies uncover significant inter-tumoral and intra-tumoral heterogeneity in tumor immune profiles, diverse immunosuppressive populations, less-defined immune cell subsets, and signal transduction networks in related cancer types. However, such analyses have not been applied to ESCC.

To describe the immune landscape of ESCC, we used high-dimensional scRNA-seq to total immune cells isolated from seven surgically removed ESCC tumors and their matched adjacent tissues. T cell receptor (TCR) sequencing was also conducted to retrieve information on T cell clonality. Our analyses revealed inter-tumoral heterogeneity among individual ESCC patients. A subgroup of ESCC tumors presented significantly increased infiltration and clonal expansion of T cells, compared with their matched adjacent tissues. However, we identified exhausted T cells, exhausted NK cells, regulatory T (Treg) cells, alternatively activated macrophages (M2), and tolerogenic dendritic cells (tDCs) in these tumors, indicating an inflamed but immune-suppressed TME in ESCC. Transcriptional profile coupled TCR-sequencing revealed lineage connections in CD4 and CD8 T cells populations. We discovered exhausted CD8 T cells showing continuous progression from a pre-exhausted state to an exhausted state. Exhausted CD4, CD8 T, and NK cells were major proliferative cell components in the TME. Additionally, we identified crosstalk among macrophages and Tregs through ligand–receptor interactions that may contribute to the immune suppressive state and disease progression. Furthermore, we identified a gene signature that was significantly associated with the survival of patients with ESCC. Our results comprehensively characterized tumor-infiltrating immune cells, revealed the landscape of the suppressive immune state, and set the baseline for applying and developing immunotherapies for ESCC.

## Results

**scRNA-seq of immune cells isolated from ESCC.** To generate a deep transcriptional map of immune cells in human ESCC, we profiled single-cell gene expression programs and coupled TCR-sequencing from CD45+ cells infiltrating immune cells isolated from seven pairs of fresh, surgically removed tumors and matched adjacent tissues of ESCC (Fig. 1a). The clinical information and hematoxylin–eosin (HE) results staining from analyzed samples were shown in Supplementary Table 1 and Supplementary Fig. 1. After removing low-quality cells, a total of 80,787 cells (3248–9097 per sample) were retained for further analysis. In these cells, a median of 1170 genes per cell was detected.

To enable a systematic analysis of immune cell populations, we normalized and pooled single-cell data from all samples and conducted unsupervised clustering to identify distinguishable populations. The whole procedure was performed using Seurat v3.0 with default parameters[17]. We annotated these populations using their canonical markers and successfully identified the major types of tumor-infiltrating immune cells as shown in other cancers, including T cells, NK cells, monocytes/macrophages, dendritic cells (DCs), B cells, plasma cells, and mast cells, as well as a very small fraction (1.31%) of other non-immune cells that were mixed in with the sorted cells (Fig. 1b). The expression of classic markers of these cell types was consistent with the annotation (Fig. 1c, d). We then analyzed "other" cluster form tumors, and found that most cells had copy number variations (CNVs), including both amplifications and deletions, suggesting that this cluster included tumor cells (Supplementary Fig. 2)

By comparing the percentages of each cell type in CD45+ cells between tumor and adjacent tissues, we found an increase of T cells and monocytes/macrophages in tumors. In contrast, the percentages of B and NK cells were decreased (Fig. 1e and Supplementary Fig. 3a). In agreement with recent studies[18], we found a large degree of variation in the immune composition among tumors (Fig. 1f, g, and Supplementary Fig. 3b). T lineage cells were the most abundant immune cell type in most tumors, making up 30–71% of the total CD45+ cells (Fig. 1g). However, considering the ratios of each immune cell type to all cells analyzed by flow cytometry during CD45+ cell isolation, there was high variation between matched tumor and adjacent tissues, as well as among individuals (Supplementary Data 1). Seven pairs of samples were roughly divided into two groups. There were only minor differences between the matched adjacent and tumor tissues in three tumor-adjacent tissue pairs (S133, S134, and S150). T cells made up to fewer than the 2% of total cells in these tumors. In contrast, the immune profiles of four other tumor-adjacent pairs (S135, S149, S158, S159) presented a significant shift in a PCA, in which 6–12% of total cells were T cells in tumors (Fig. 1h, i). These tumors also showed increased numbers of monocytes/macrophages, compared with other tumors and adjacent tissues (Supplementary Fig. 3c). In addition, we found inter-patient variation in biologic signatures, including hypoxia, inflammation response, and TNFA-via NFKB pathways in lymphocytes. Interestingly, S135 and S158 showed similar gene signatures enrichment, and S133 and S134 showed similar gene signatures enrichment in these pathways (Supplementary Fig. 3d–f).

Next, we further validated our results for the major immune cell types with additional samples by flow cytometry and immunohistochemistry (IHC). We found an increase in T cells and macrophages and a decrease in NK and B cells in tumors, compared to adjacent tissues, which is consistent with the scRNA-seq data (Supplementary Figs. 4 and 5). Notably, neutrophils were not identified in scRNA-seq as a population like others reported[12,18–20], but they were detected in low abundance by flow cytometry and IHC. The failure to detect neutrophils in scRNA-seq may be caused by the combination of the low abundance of neutrophils in ESCC and the limitation of the current 10× scRNA-seq technique. Neutrophils' low RNA content and abundance of RNases may lead to increased sensitivity to prolonged processes of scRNA-seq, which could

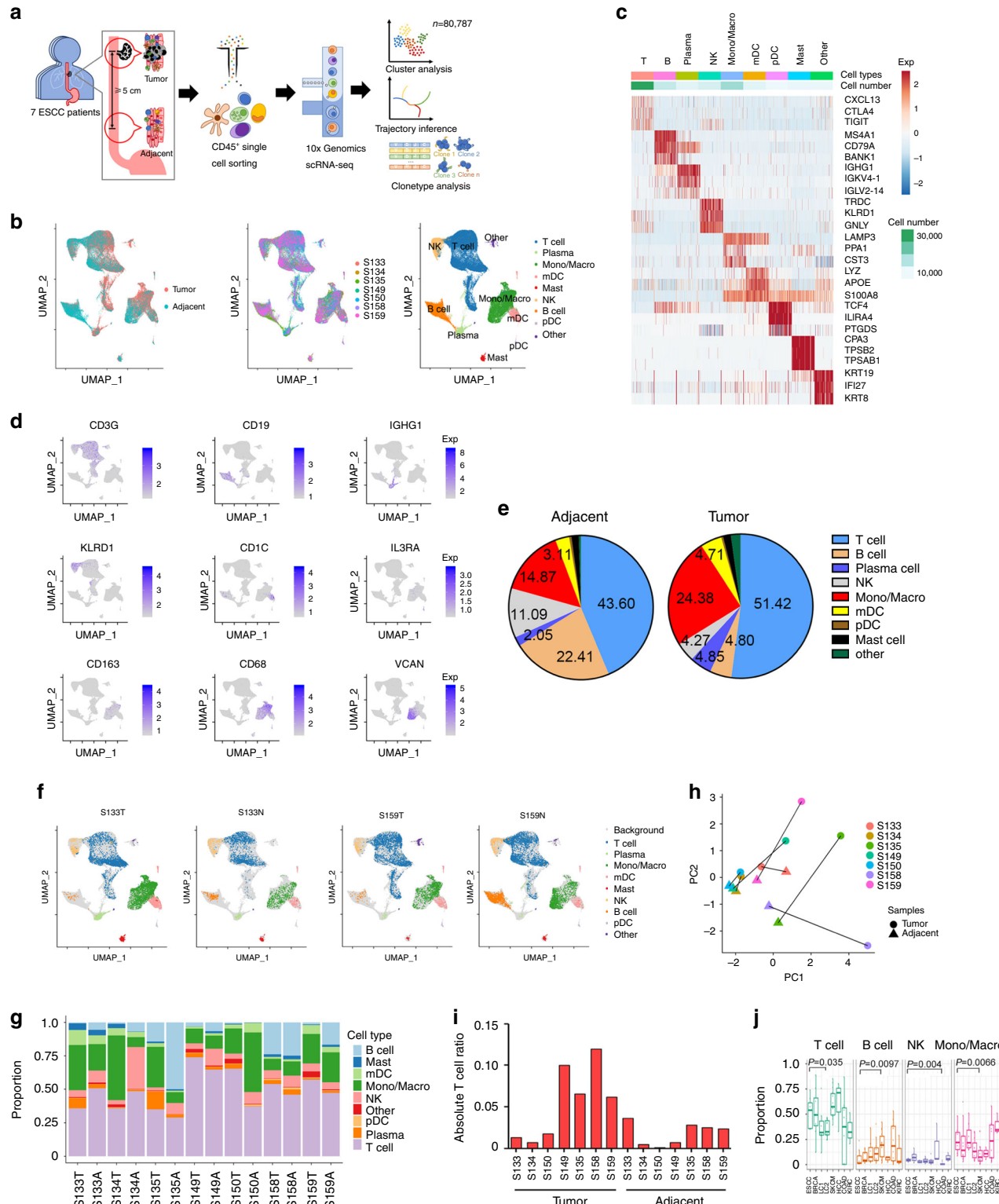

potentially result in fewer transcripts being detected, resulting in these cells not passing quality control.

Next, we compared the major compartments of infiltrating immune cells in ESCC to other cancer types with available data[7,12,18–22]. We found that ESCC was among the tumor types with a higher number of infiltrating T cells and monocytes/macrophages and a lower number of infiltrated B cells (Fig. 1j). This is consistent with our observation that ESCC had increased T cells and monocytes/macrophages and decreased B cell ratios,

compared to their adjacent tissues (Fig. 1e). Notably, recent studies suggested positive effects of tumor-infiltrating B cells, especially those in tertiary lymphoid structures, on increasing response to immunotherapies[23]. Whether it is responsible for the response of checkpoint blockade in ESCC needs an additional investigation.

**Clustering and subtype analyses of T and NK cells.** Since T and NK cells are the major cytotoxic immune cells in the TME, we

**Fig. 1 Profile of immune infiltrates in human ESCC with scRNA-seq and scTCR-seq. a** Schematic diagram of the experimental design and analysis. **b** UMAP plot of 80,787 high-quality immune cells to visualize cell-type clusters based on the expression of known marker genes. **c** Heatmap of the relative expression level of genes across cells, sorted by cell type. The expression was measured as the z-score normalized log2 (count+1). **d** Expression levels of relative marker genes across 80,787 cells illustrated as UMAP plots. The expression was measured as the log2 (count+1). **e** Pie charts of cell-type fractions for ESCC tumor and adjacent tissues' infiltrating immune cells, colored by cell type. **f** UMAP plot of complete immune systems from seven ESCC tumor and adjacent tissues, representative of S133 and S159; other samples were shown in Supplementary Fig. 3b. Cells were colored by clusters and labeled with the inferred cell types. **g** The proportion of cells that contributed to each cluster by each sample, colored by cell type. A, Adjacent; T, Tumor. **h** PCA was performed on the absolute ratio of cell types of 14 samples from seven patients to see the heterogeneity between samples, colored by sample. **i** The absolute T cell ratio in total cells for the 14 samples. **j** Comparison of the fractions of T cells, B cells, NK, monocytes/macrophages in tumors from patients with ESCC ($n = 7$), BRAC ($n = 8$), LC1 ($n = 7$), LC2 ($n = 11$), SKCM ($n = 16$), HCC ($n = 3$), COAD ($n = 10$),n and KIRC ($n = 3$). Skin cutaneous melanoma (SKCM); breast invasive carcinoma (BRCA); lung cancer (LC); hepatocellular carcinoma (HCC); colon cancer. (COAD); kidney cancer (KIRC). Each box represented the interquartile range (IQR, the range between the 25th and 75th percentiles) with the mid-point of the data, and whiskers indicate the upper and lower values within 1.5 times the IQR. P value was calculated by two-tailed Wilcoxon sum rank test.

conducted unsupervised clustering of T and NK cells that were pooled from all samples. We identified six CD4 T clusters, seven CD8 T clusters, one CD4 and CD8 double negative T cells cluster and three NK clusters (Fig. 2a). The top differentially expressed genes of each cluster are shown in Supplementary Data 2. Among T cells, we used known functional markers to suggest CD4 T cell populations, including naïve, memory, effector, exhausted T cells, and Tregs. The markers also identified CD8 T cell populations, including memory, effector, cytotoxic, and exhausted T cells (Fig. 2b). CD4-C1-CCR7 carried a naïve signature, including TCF7, CCR7, LEF1, and SELL, and expressed very low levels of cytokines and effective genes. CD4-C6-FOXP3 expressed high levels of Treg signature genes FOXP3, IKZF2, IL2RA, and CTLA-4, as well as co-stimulatory markers, such as CD28, ICOS, TNFRSF9, and TNFRSF14. CD4-C5-STMN1 cells expressed high levels of CD38, ENTPD1, TNF, and HIF1A that were distinguished from CD4-C6-FOXP3 (Fig. 2b). Among CD8 T cells, three clusters (CD8-C5-CCL5, CD8-C6-STMN1, CD8-C7-TIGIT) expressed variable levels of checkpoint molecule genes, including PDCD1, TIGIT, CTLA-4, HAVCR2, and LAG-3, representing the phenotypes of exhausted cells. These cells highly expressed CD38, CD39 (ENTPD1), and CD103 (ITGAE), which also displayed an exhausted tissue-resident memory phenotype[24], as well. Interestingly, most cytotoxic markers were also highly expressed in exhausted CD8 T cells, such as IFNG and GZMB, except for TNF and IL2, which is consistent with observations made by other reports[14,25]. Another cluster (CD8-C1-NKG7), which expressed high levels of granzyme genes and NKG7 but the lowest level of checkpoint molecule genes and SELL, TCF7, was likely the recently activated effector T cells (TEMRA)[14] (Fig. 2b). Similar to a recent report[7], we identified CD8-C3-GZMK as a transitional population that presented a distinct expression pattern of transcription factors, compared with other CD8 clusters, highlighted by a very high level of EOMES. Interestingly, this cluster was also the only one to have cells expressing a high level of GZMK, which suggested a relationship between EOMES and GZMK. Indeed, we found that GZMK and EOMES had a positive correlation in both CD8-C3-GZMK and total CD8 T cells (Supplementary Fig. 6a, b).

To investigate gene networks in cytotoxic and exhausted CD8 T cells and Treg cells, we used the public naïve, Treg, exhaustion, and cytotoxic signatures[10] (Supplementary Data 3) and applied these signatures to CD8 and CD4 clusters and computed a transcriptional score. In CD4 T cells, CD4-C6-FOXP3 had the strongest Treg signature, while CD4-C5-STMN1 had the strongest exhaustion signature; CD4-C4-IFIT3 was enriched in cytotoxic signature (Fig. 2c). Consistent with previous analyses, CD8-C1-NKG7 was the most active cytotoxic CD8 T cells, whereas CD8-C5-CCL5 and CD8-C6-STMN1 had lower exhaustion scores than CD8-C7-TIGIT (Fig. 2d). The naïve score was very low in CD8 T

cell clusters, which is consistent with Fig. 2b; no naïve clusters of CD8 T cells were identified. This result suggested that most of the tumor-infiltrating CD8 T cells were in the active, memory, or exhausted states in ESCC (Fig. 2d). We further analyzed the genes whose expression was highly correlated with the expression of FGFBP2, LAG3, and FOXP3[7]. The top 50 genes were then used as signatures for cytotoxicity, exhaustion, and Treg with the top 30 genes shown in Supplementary Fig. 6c. We then use these signatures to analyze T cells clusters and found that the enrichment scores were consistent with the published signatures (Supplementary Fig. 6d, e).

There were lineage connections within CD4 populations. Some of the genes activated in Tregs (CD4-C6-FOXP3) overlapped with genes characteristic of the exhaustion program in CD4 T cells (CD4-C5-STMN1). We compared the gene expression of both clusters with naïve-like cell population (CD4-C1-CCR7). Genes enriched in both exhausted and Treg cells included regulatory molecules and many co-inhibitory and co-stimulatory receptors, such as TNFRSF9, CSF1, and TIGIT. In contrast, CD4-C6-FOXP3 expressed much higher levels of FOXP3, IL2RA, and CTLA4 than CD4-C5-STMN1, while CD4-C5-STMN1 expressed higher levels of CCL5, CCL4, IFI6, TOX, PDCD1, CXCL13, IFNG, and ID2 than CD4-C6-FOXP3 (Fig. 2e and Supplementary Fig. 6f). Visualization of the exhaustion and Treg scores confirmed the overlap between these two clusters (Fig. 2c). Cytotoxic and exhausted CD8 T cells both expressed many effector molecules such as GNLY and GZMH, while exhausted CD8 T cells expressed a higher level of IFNG than cytotoxic cells (Fig. 2f), which suggested that exhausted T cells still expressed high levels of some effector molecules and tried to respond to tumor cells. Different stages of CD8 T cell exhaustion have been reported in other cancer types[26]; pre-exhausted T cells expressed intermediate levels of PD1, EOMES, TOX, and NFAT2, while exhausted T cells expressed high levels of these molecules. We found that CD8-C5-CCL5 expressed lower levels of PDCD1 and EOMES than CD8-C7-TIGIT, while CD8-C6-STMN1 was in the middle (Fig. 2g). We also checked the expression levels of TOX and NFATC2; both genes participate in establishing epigenetic programs to install permanent exhaustion status in CD8 T cells[27,28]. The data showed that CD8-C7-TIGIT expressed high levels of TOX and NFATC2 (Fig. 2g). This suggests that CD8-C5-CCL5 were at an early stage of exhaustion, while CD8-C7-TIGIT was in the exhaustion stage and CD8-C6-STMN1 was likely a transition stage between CD8-C5-CCL5 and CD8-C7-TIGIT.

Dr. Simoni and colleagues reported that large fractions of tumor-infiltrating CD8 T cells are bystanders that recognize cancer unrelated epitopes. These cells lack CD39 and are phenotypically distinct from tumor antigen-specific CD8 T cells[29]. We then analyzed CD39 expression in CD8 T cells in

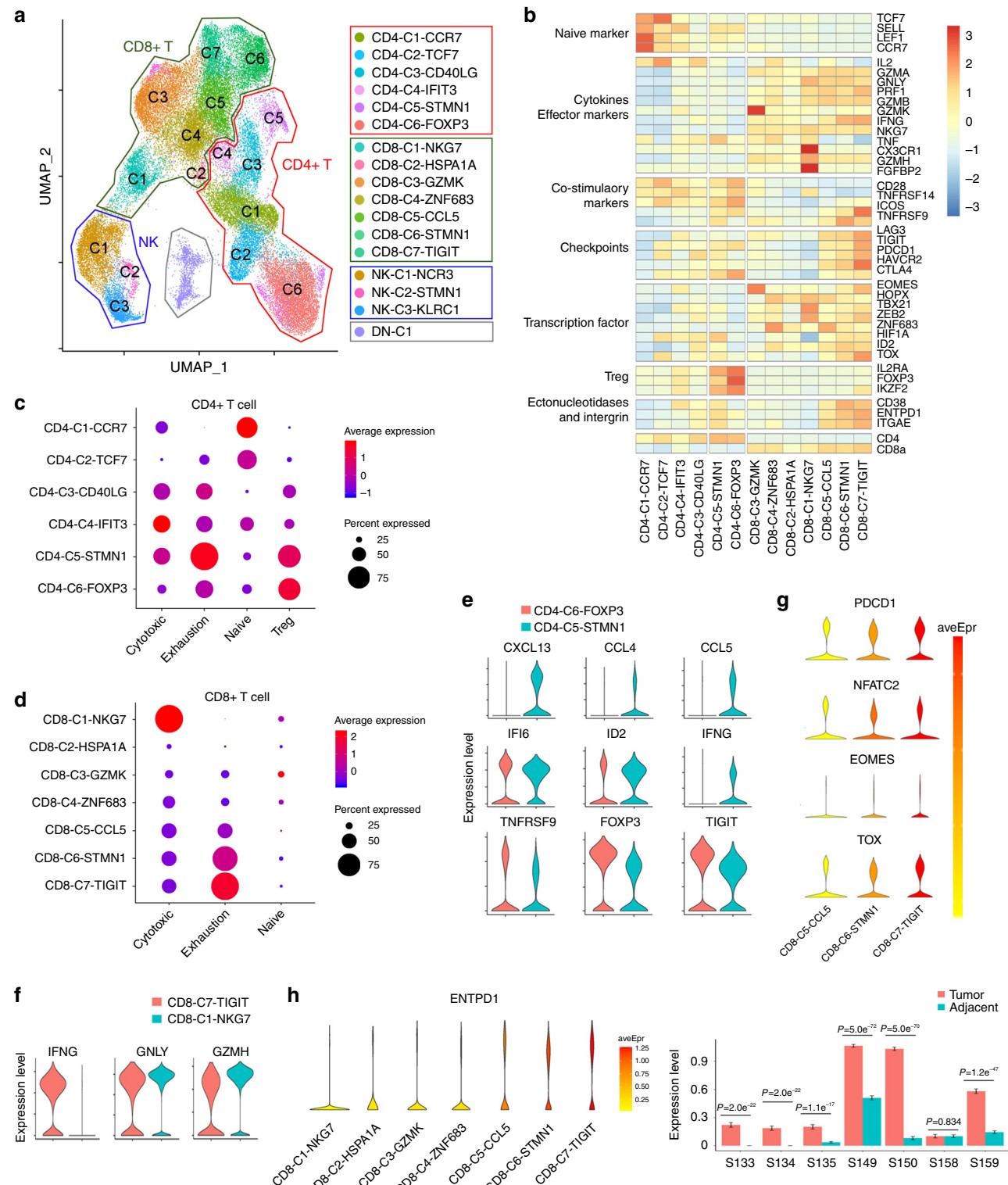

**Fig. 2 Detailed characterization of T cells in ESCC. a** UMAP plot of 44,634 single T and NK cells from 14 samples, showing the formation of 17 main clusters, including six for CD4 T cells, seven for CD8 T cells, one for CD4 and CD8 negative T cells cluster, and three for NK cells. Each dot corresponds to a single cell, colored according to the cell cluster. **b** Heatmap of Z-score normalized log2 (count+1) expression of selected T cell function-associated genes in each cell cluster. **c** Dot plot of representative cytotoxic, exhaustive, naïve, and Treg signatures in CD4 T cell clusters, Z-score normalized log2 (count+1). **d** Dot plot of representative cytotoxic, exhaustive, and naïve signatures in CD8 T cell clusters, Z-score normalized log2 (count+1). **e, f** Violin plot showing the *CXCL13, CCL4, CCL5, IFI6, ID2, IFNG, TNFRSF9, FOXP3,* and *TIGIT* in CD4-C6-FOXP3 and CD4-C5-STMN1 cells **e** *IFNG, GNLY,* and *GZMH* in CD8-C7-TIGIT and CD8-C1-NKG7 cells **f**. The expression was measured as the log2 (count+1). **g** Violin plot showing *PDCD1, NFATC2, EOMES,* and *TOX* in the CD8-C5-CCL5, and CD8-C6-STMN1, CD8-C7-TIGIT clusters. The expression was measured as the log2 (count+1). **h** Violin plot showing *ENTPD1* expression in CD8+ T cell clusters from 7 paired ESCC and adjacent samples. The expression was measured as the log2 (count+1). Data are presented as mean ± SD; P value was calculated by two-tailed Student's t-test.

our samples. The results showed that CD39 expression was significantly higher in pre-exhausted and exhausted CD8 T cells (C5, C6, C7), compared to other clusters. Additionally, most CD8 T cells expressed higher CD39 in tumors than adjacent tissues except S158 (Fig. 2h). CD39-CD8 T cells may be bystander cells and recognize non-tumoral antigens, such as Epstein-Barr virus, human cytomegalovirus, or influenza virus, which are commonly found in the esophagus.

**Altered status of T and NK cells in tumors**. We compared T cell clusters between tumors and adjacent tissues. The percentages of Treg cluster CD4-C6-FOXP3 and exhausted CD4 T cells CD4-C5-STMN1 in CD45+ cells were significantly increased in tumors compared with matched adjacent tissues (Fig. 3a, b). Indeed, Tregs and exhausted CD4 T cells were more than 50% of the total CD4 T cells in tumors, while they were only 25% in adjacent tissues (Fig. 3a, b). Flow cytometry also demonstrated the enrichment of Tregs in ESCC tumors (Fig. 3c). Similarly, exhausted CD8 T cells were enriched in tumors. The total percentage of exhausted CD8 T cells was <20% in adjacent tissues, but 57% in tumors (Fig. 3d, e). Consistently, PD1 expression in CD8 T cells was higher in ESCC (Fig. 3f). In contrast, the most active cytotoxic CD8 T cell group (CD8-C1-NKG7) significantly decreased, from 23% in adjacent tissues to 4% in tumor tissues on average (Fig. 3e). The significant increase in Tregs and exhausted CD4 and CD8 T cells in tumor tissues indicated an immune suppressive environment. Analyzing matched tumor-adjacent tissues produced similar results (Supplementary Fig. 7a–f).

We also observed a substantial decrease in NK cells in tumor tissues, compared with matched adjacent tissues (Fig. 1e). Additionally, the major cluster of NK cells switched from NK-C1-NCR3 in adjacent tissues to NK-C3-KLRC1 in tumors, and NK-C2-STMN1 also increased dramatically in tumors (Fig. 3g). NK-C1-NCR3 expressed high levels of NCR3, CD266, NKG7, and LAMP1 (Fig. 3h). In contrast, NK-C3-KLRC1 and NK-C2-STMN1 clusters expressed KLRC1 and ITGA1 inhibitory receptors at high levels (Fig. 3h). Flow cytometry assays verified increased NKG2A (KLRC1) expression in NK cells in ESCC compared with adjacent tissues (Fig. 3i). Indeed, NK-C3-KLRC1 and NK-C2-STMN1 had extremely low-cytotoxic scores; in contrast, the exhaustion scores were elevated (Fig. 3j), which indicated that NK cells were insufficient and function impaired in ESCC.

We further analyzed the cell cycle in T and NK cell clusters to determine the proliferating ability of cells. We generated a proliferation score of cell cycle genes that were previously shown to denote G1/S or G2/M phases[30,31] and used it to infer the proliferation status of T and NK cell clusters. Interestingly, we found that CD4-C5-STMN1, CD8-C6-STMN1, and NK-C2-STMN1, which were enriched in exhaustion genes, were all highly proliferative (Fig. 3k–m), which is consistent with recent research suggesting that exhausted T cells are the major intra-tumoral proliferating immune cell compartment[7].

**Clonality of CD4 and CD8 T cells**. To determine whether the clonal selection and amplification of T cells contributed to the observed phenotypic diversity, we further analyzed the results from the coupled TCR sequencing from the same samples. We recovered TCRα and TCRβ sequences from 26,920 and 31,440 T cells, respectively. The percentages of unique and productive α chains and β chains were 70.59% (26,920/38,134) and 82.45% (31,440/38,134), respectively. The proportion of T cells that had both chains was 69.94% (26672/38134), in accordance with previous reports[32]. We observed a total of 15,654 unique TCR sequences. Clonal expansion was observed, with clonal sizes

ranging from 2 to 2600 (Fig. 4a). No shared clones were found between patients, as expected. Consistent with studies from other cancer types, the majority of TCRs were unique. However, TCR clonotype composition was highly variable among patients. While some patients showed minimal clonal expansion (S134, S135, and S158), others were strongly dominated by a small number of T cell clones (S149 and S150). Indeed, S149 and S150 tumors showed 65% and 68% of T cells with TCRs shared by more than two cells, indicating the high clonal expansion of T cells in these tumors (Fig. 4b). Four out of the seven patients' tumors had an increase in expanded clones, compared with the matched adjacent tissue (Fig. 4b). Furthermore, each cluster was, in fact, composed of different combinatorial subsets of the clonotypes (Fig. 4c and Supplementary Fig. 8a). CD8 T cells had significantly more clonal cells than CD4 T cells in general, and the naïve cluster CD4-C1-CCR7 displayed very limited clonal expansion (Fig. 4d). CD8-C1-NKG7, the cytotoxic cluster in CD8 T cells, which had a higher frequency in adjacent tissues, also showed increased clonal expansion in adjacent tissues than in tumor tissues (Fig. 4e). However, the Tregs in tumors had an increased number of clones compared to match-adjacent tissues (Fig. 4f), suggesting that the expansion of specific clone cells may be responsible for the higher percentage of Tregs in tumors. While most cells contained unique TCRs, clonal amplification was observed to varying degrees in different clusters. Indeed, we found sharing of TCR sequences among all clusters in CD4 cells, including Tregs, and all clusters within CD8 cells, with the exception of C2 (Fig. 4g, h). The number of clones shared between CD8-C7-TIGIT and CD8-C5-CCL5 and CD8-C6-STMN1 was 166 (9.0%) and 156 (8.4%), respectively (Fig. 4i). Interestingly, CD8-C7-TIGIT cells in the adjacent tissues shared more clonotypes with other CD8 clusters (Supplementary Fig. 8b). CD4-C6-FOXP3, the Treg cluster, had the same trend in tumors, displaying 14.4% of shared clonotypes with CD4-C1-CCR7 and 40.7% in the adjacent tissue (Fig. 4j and Supplementary Fig. 8c). However, clonal T cells in cytotoxic, exhausted, and Treg cells shared limited TCR between the tumor and adjacent tissues (Supplementary Fig. 8d), which suggested potential common origins of some Tregs and naïve CD4 T cells. Our data are consistent with a recent study[10] reporting that T cells of different clusters are not completely independent, but might undergo an extensive state transition.

**Distinct functional composition of myeloid cells in ESCC**. Next, we conducted unsupervised clustering of myeloid cells. Fourteen clusters were identified, including nine clusters of monocytes/macrophages and five clusters of DCs (Fig. 5a). The top differentially expressed genes are shown in Supplementary Data 4. A heat map showed the cluster gene signatures and the gradient development between monocytes/macrophage clusters (Supplementary Fig. 9a). Using published gene signatures of monocytes, classically activated macrophages (M1), alternatively activated macrophages (M2), and myeloid-derived suppressor cells (MDSCs)[18,33] (Supplementary Data 5), macrophage clusters were identified, namely Macro-C1-IL6, representing the M1; Macro-C3-CSF1, representing the M2; MDSC-C1-C1QC and MDSC-C2-APOE representing MDSC[34,35], Macro-C2-IL1RN may represent the intermediate state, both enriched monocyte, M1 and M2 signature, Macro-C4-LILRB2, preferential and enrichment in adjacent mucosa versus tumors were denoted as tissue resident macrophages (TRM); Mono-C1-VCAN showed a strong monocyte signature (Fig. 5b). To further understand the cell transitions, we used Monocle[36], an unsupervised inference method, to construct the potential developmental trajectories of cell conversion. Data showed that monocyte (Mono-C1-VCAN), M1 (Macro-C1-IL6), and M2 (Macro-C3-CSF1) were at the end

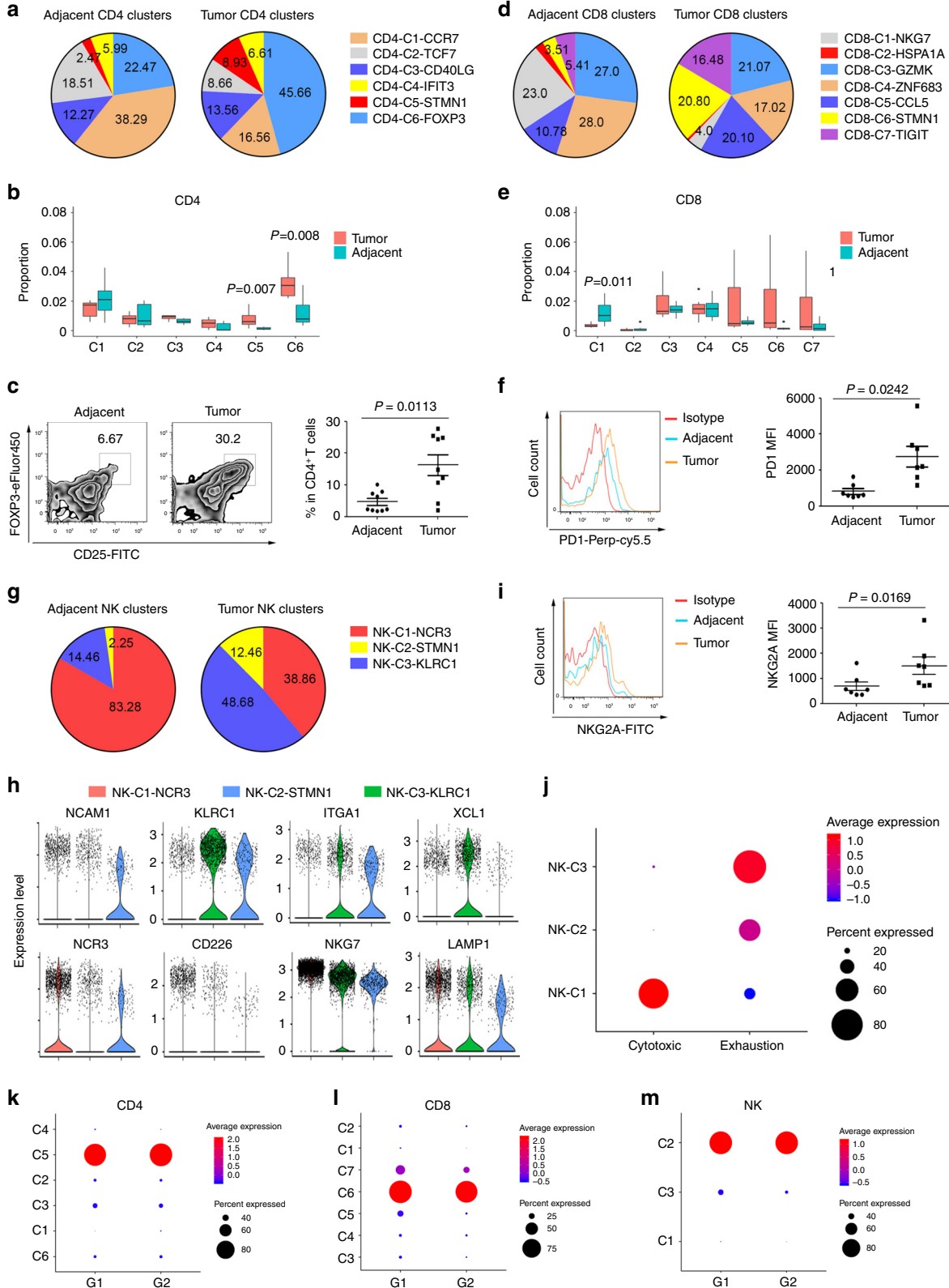

of branches, and Macro-C2-IL1RN was located in the middle (Fig. 5c). We further applied another algorithm Slingshot[37] to the same dataset and obtained comparable results (Supplementary Fig. 9b). However, we also observed a significant correlation between M1 and M2 signatures in macrophages (Supplementary Fig. 9c), which indicated a complicated macrophage polarization process in ESCC, which was consistent with other researches[12,18].

Interestingly, the monocyte (Mono-C1-VCAN) and TRM (Macro-C4-LILRB2) were significantly more redundant in adjacent tissues. In contrast, M2 (Macro-C3-CSF1) and MDSC (MDSC-C1-C1QC) were enriched in tumors (Supplementary Fig. 9d). Flow cytometry data showed that CD68+ macrophages expressed increased levels of CD163 and CD206 in ESCC compared with adjacent tissue (Supplementary Fig. 9e). These results

**Fig. 3 Altered status of T and NK cells in tumors. a** Pie charts of CD4 T cell cluster fractions for ESCC tumors and adjacent tissues, colored by cluster type. **b** Proportions of clusters of CD4 T cells in the tumors and adjacent tissues ($n = 7$). Each box represents the interquartile range (IQR, the range between the 25th and 75th percentile) with the mid-point of the data, whiskers denote 1.5 times the IQR. Two-tailed paired Student's $t$-test. **c** Proportions of CD4+CD25hiFOXP3+ Tregs in ESCC and adjacent tissues ($n = 9$), Data are presented as mean ± SEM; $P$ value was calculated by two-tailed paired Student's $t$-test. **d** Pie charts of CD8 T cell cluster fractions for ESCC tumors and adjacent tissues, colored by cluster type. **e** Proportions of clusters of CD8 T cells in tumor and adjacent tissues ($n = 7$). Each box represents the interquartile range (IQR, the range between the 25th and 75th percentiles) with the mid-point of the data; whiskers indicate the upper and lower values within 1.5 times the IQR. Two-tailed paired Student's $t$-test. **f** Flow cytometry measured CD8 T cell PD1 expression in ESCC and adjacent tissues ($n = 7$). Data are presented as mean ± SEM; $P$ value was calculated by two-tailed paired Student's $t$-test. **g** Pie charts of NK cell cluster fractions for ESCC tumor and adjacent tissues, colored by cluster type. **h** Violin plots comparing the indicated gene expression in NK cell clusters. The expression was measured as the log2 (count + 1). **i** Flow cytometry measured NK cells NKG2A expression in ESCC and adjacent tissues ($n = 7$). Data are presented as mean ± SEM; $P$ value was calculated by two-tailed paired Student's $t$-test. **j** Dot plot of cytotoxic and exhaustion signatures in NK clusters. $Z$-score normalized log2 (count + 1). **k–m** Dot plot of proliferation signatures in CD4 **k**, CD8 **l**, and NK **m** clusters. $Z$-score normalized log2 (count + 1).

were consistent with the immune-suppressive function of tumor-associated macrophages (TAMs) in ESCC. Next, we used WGCNA to conduct weighted-correlation network analysis in monocytes/macrophages. Interestingly, we found that the Turquoise module was positively correlated with the monocyte clusters, Mono-C1-VCAN and Mono-C2-IL1B, and negatively correlated with the M2 cluster Macro-C3-CSF1 and MDSC clusters MDSC-C1-C1QC, MDSC-C2-APOE (Fig. 5d, e). The genes in this module were associated with myeloid leukocyte activation, activation of immune response (Supplementary Fig. 9f). We further analyzed the genes in this module and their association with Mono-C1-VCAN (Fig. 5f) to select the top 50 genes that most correlated to form a signature set (Supplementary Data 6). Interestingly, this signature was strongly associated with a high probability of progression-free survival in ESCC (Fig. 5g), as well as in cervical squamous cell carcinoma and lung squamous cell carcinoma (Supplementary Fig. 9g), which suggests that this signature may serve as a prognostic biomarker in ESCC and squamous cell carcinoma in other tissues.

We further applied single-cell regulatory network inference and clustering (SCENIC) method[38] to explore the transcription factors that may regulate monocyte, M1, and M2, development. MITF, BHLHE40, ATF3, and USF2 were upregulated in M2, whereas IRF transcription factors, including IRF1, IRF7, IRF2, IRF5, and PRDM1 were upregulated in M1. RARA, FOSB, and NFKB2 were greatly increased in the monocyte clusters (Fig. 5h and Supplementary Fig. 9h). Strikingly, SCENIC also revealed a dichotomy between most tumor and adjacent tissue pairs, except S158 (Fig. 5i). The transcriptional factor BHLHE40 was specifically expressed in M2. We conducted the network analyses to identify the BHLHE40 downstream genes and analyzed the functions through Metascape (Fig. 5j). Data showed that BHLHE40 downstream genes were associated with myeloid cell differentiation, and negatively regulated the cellular response (Supplementary Fig. 9i). BHLHE40 has recently been reported to mediate tissue-specific control of macrophage self-renewal, and proliferation[39] and we found that BHELHE40 was associated with poor prognosis of ESCC (Supplementary Fig. 9j). BHLHE40 may play a critical role in inducing TAMs toward the M2 phenotype, and further studies are needed to explore the detailed mechanisms. Our data suggested that an enrichment of suppressive TAMs in the ESCC microenvironment may contributed to the progression of disease, and they elucidated some compelling TF candidates associated with prognosis.

Five DC clusters featured high expression levels of CLEC9A, CD1C, FCER1A, LAMP3, and CLEC4C, consisting of conventional cDC1 (DC-C1-CLEC9A), cDC2 (DC-C2-CLEC10A), monocyte-derived DC (DC-C4-FCER1A), LAMP3 + DC (DC-C3-LAMP3), and pDC (DC-C5-CLEC4C) (Fig. 6a). DC-C3-LAMP3 was enriched in tumors, compared to adjacent tissues (Fig. 6b). Zhang Q, et al. recently reported that LAMP3 + DCs

were the most activated DC subset with potential migration capacity in tumors and that they may originate from both cDC1 and cDC2[12]. Indeed, we compared the activation and migration scores of DCs and found that LAMP3 + DCs had the highest activity and migration ability compared to other DC subsets (Fig. 6c). Strikingly, we also found that LAMP3 + DCs enriched the tolerogenic signature (Fig. 6c), which was described in a previous study[40]. LAMP3 + DCs expressed many regulatory molecules, such as IDO1, EBI3, CD274, and IL10 (Fig. 6d). When conducting pathway-enrichment analysis, we found that genes upregulated in LAMP3 + DCs were enriched in the pathways of cytokine-mediated signaling transduction, DC cell differentiation, leukocyte activation, membrane trafficking, antigen processing, and presentation (Supplementary Fig. 10a), which supported this DC subset as multifunctional. We then verified LAMP3 + DCs by flow cytometry. The data showed that LAMP3 + DCs expressed significantly higher CD83, CCR7, and PDL1 than LAMP3-DCs (Fig. 6e, f), suggesting the maturation, migration, and regulation ability of LAMP3 + DCs. Multi-color IHC staining also validated the existence of CD11C + LAMP3 + PDL1 + IDO + DCs in tumor tissue (Fig. 6g). We further treated DCs with IFNγ and LPS. Interestingly, we found that IFNγ and LPS stimulation-induced DCs expressing PDL1 and IDO (Fig. 6h) and had an increased ability to induce FOXP3 expression when co-cultured with CD4 + CD45RA + naïve T cells (Fig. 6i). These data suggested that IFNγ and LPS may induce the tolerogenic DCs in vitro. Furthermore, SCENIC analysis revealed that DC subsets could be distinguished by different groups of transcription factors (Fig. 6j). LAMP3 + DCs showed higher levels of RELB, IRF1, FOXO1, and ETS1; and CEBPD, ETS2, CEBPB, CREB5 were upregulated in cDC2. BCL6, BACH1, FLI1, and RUNX1 were highly expressed in cDC1, while high levels of SPIB, IRF7, and NR3C1 were associated with pDCs (Fig. 6j and Supplementary Fig. 10b). RELB has been reported to regulate cDC development by hematopoietic extrinsic mechanisms[41]. A new RELB-dependent CD117 + CD172a + murine DC subset preferentially induces Th2 differentiation and supports airway hyperresponses in vivo[42]. We thus conducted network analyses to identify the RELB downstream genes through Metascape. The data showed that these genes were associated with cell migration, DC differentiation, and negative regulation of cellular processes (Supplementary Fig. 10c, d). These results suggested an important role of RELB in regulating tDC development.

**Cell–cell interaction between immune cells in ESCC.** Cell–cell communications, mainly through ligand–receptor interactions, play key roles in determining the TME and responding to therapeutics. Thus, we performed systematic analyses on the potential cell–cell interactions based on co-expression of known ligand–receptor pairs in any two types of tumor-infiltrating immune cells[43] and compared them between tumor and adjacent

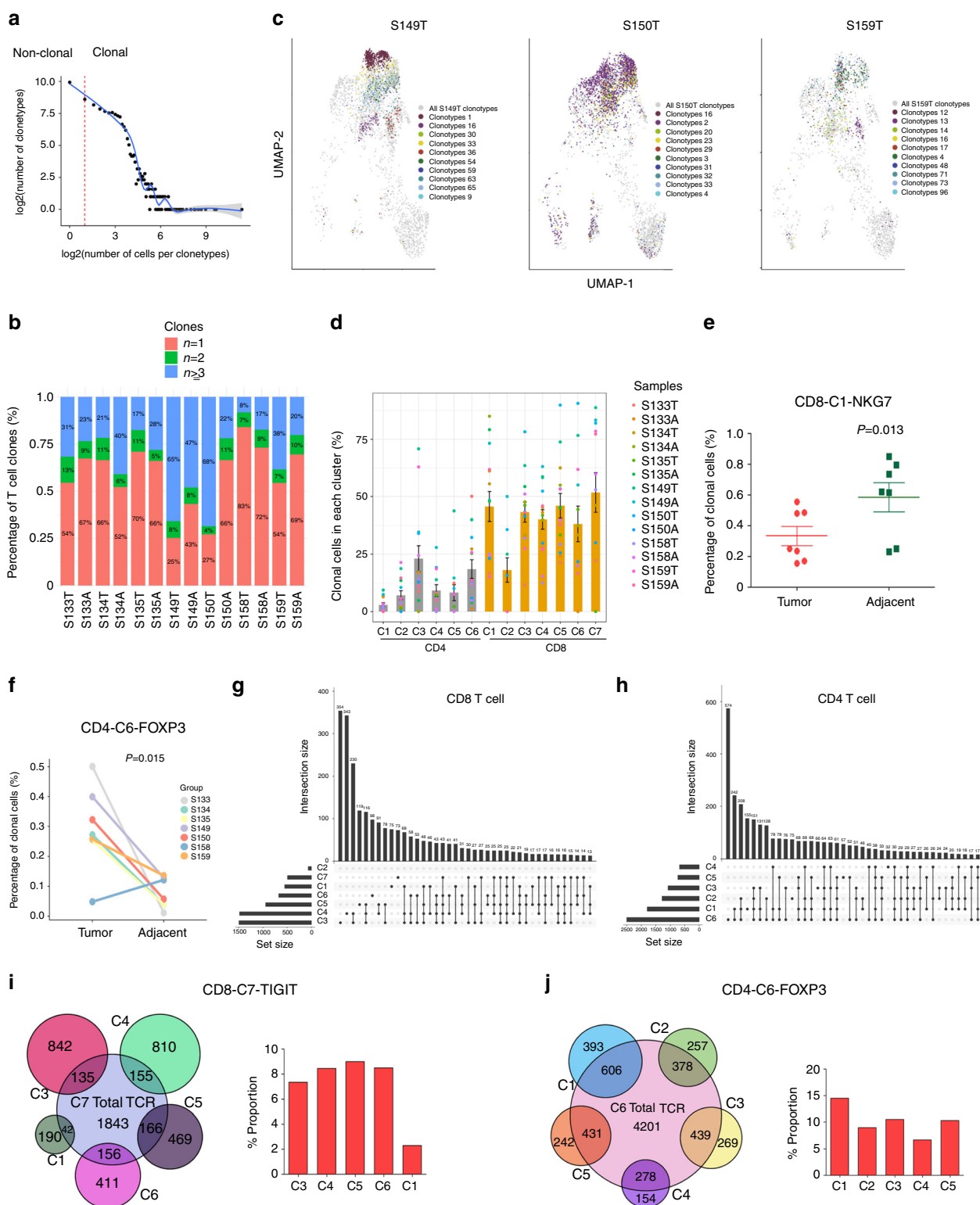

tissues. Because of the high infiltration levels and the important immune regulatory roles of macrophage and Tregs in tumors, we applied scTHI (https://github.com/miccec/scTHI), another widely used method to analyze macrophages and Tregs interactions and found that both of them had comparable results (Supplementary Data 7).

We found that the interaction of TNF-TNFSF1B, CCL4-CCR8, and IL-1β−IL1R2 between macrophages and Tregs had a high interaction score and Tregs expressed high levels of TNFSF1B,

CCR8, and IL1R2 in tumors (Fig. 7a–c). It has been reported that CCL4 plays a key role in the recruitment of Tregs in ESCC[44] through its receptor CCR8. TNF receptors in Tregs play important roles in sensing and dampening local inflammation[45]. However, the role of IL1R2 in regulating Treg function has not been clearly demonstrated. First, we analyzed IL1R2 expression from ESCC and adjacent tissue by flow cytometry, and found that IL1R2 expression was higher in Tregs isolated from tumors, than that in adjacent tissues (Supplementary Fig 11a). Then, multi-

**Fig. 4 Clonality of CD4 and CD8 T cells. a** The association between the number of T cell clonotypes and the number of cells per clonotype. The dashed line separates non-clonal and clonal cells; with the latter being identified by repeated usage of TCRs. Solid line LOESS fitting shows the correlation between the two axes. **b** The TCR distribution of T cells across different samples. Unique ($n = 1$), and clonal ($n = 2$, $n \geq 3$) TCRs are labeled with different colors. A, Adjacent; T, Tumor. **c** Representative examples of dominant clonotypes (top 10 in color) from each tumor (gray) identified by TCR sequencing; other samples are shown in Supplementary Fig. 6a. **d** Bar plot showing the average fractions of clone T cells among all 14 samples in each cluster, data is presented as mean ± SEM. Dot colors represent different samples. A, Adjacent; T, Tumor. **e, f** Percentage of clonal cells in CD8-C1-NKG7 **e** and CD4-C6-FOXP3 **f** from tumor and adjacent tissues ($n = 7$). Data are presented as mean ± SEM, *P* value was calculated by two-tailed paired Student's *t*-test. **g, h** Share TCR clone types between different clusters in CD4 T cells **g** and CD8 T cells **h**. The lines connecting the dots indicate the clusters sharing TCR clone types, the bar plot shows the shared TCR clone type number, and dot only suggests the unique clone types. **i** The number (left panel) and percentage (right panel) of shared TCRs between CD8-C7-TIGIT and other CD8 clusters in tumors. **j** The number (left panel) and percentage (right panel) of shared TCRs between CD4-C6-FOXP3 and other CD4 clusters in tumors.

color IHC also validated the IL1R2 expression in Tregs (Fig. 7d). Third, in vitro co-culture and antibody-blocking assays showed that IL1R2 was required for Tregs to inhibit IL-1β-dependent activation of effector T cells, such as proliferation and IFNγ expression (Fig. 7e, f, Supplementary Fig. 11b, c).

We also predicted an interaction between MHC in Tregs and LILRB1 in macrophages (Fig. 7g). The MHC receptor LILRB1 is a negative regulator of myeloid cell activation and a promotor of the M2 suppressive state. The MHC–LILRB1 interaction suppresses macrophages and is a target of cancer immunotherapy[46]. We first analyzed LILRB1 expression in macrophages by scRNA-seq and further validated it by FACS. We found that the expression of LILRB1 in macrophages increased in ESCC, compared to adjacent tissues (Fig. 7h, i and Supplementary Fig. 11d). Tregs expressed high levels of HLA-A, HLA-B, and HLA-C (Supplementary Fig. 11e, f). Multi-color IHC staining also showed potential physical interaction (co-localization) between LILRB1 expression macrophages and Tregs (Fig. 7j). When Tregs co-cultured with macrophages, we found that Tregs promoted macrophages expressing M2 markers, including CD163 and PDL1, and decreased TNFα expression. However, LILRB1 antibody blockade suppressed these affects (Fig. 7k, l and Supplementary Fig. 11g, h). These data suggested that Tregs may modulate macrophage function through HLA and LILRB1 interactions, and blocking this pathway may promote antitumor immunity in ESCC.

## Discussion

Here, we present a comprehensive characterization of immune cells in seven pairs ESCC tumors and matched adjacent tissues. Immune signature profiling and TCR β-chain repertoire analysis have been studied in ESCC using mRNA microarray and bulk RNA-seq, respectively[47,48]. Recently, a mouse model mimicking human ESCC development and construction of a single-cell ESCC developmental atlas have been reported[49]. Here, we combined deep sc-RNA-seq and TCR-seq, and illustrated the whole immune landscape, including the innate and adaptive immune cell atlas in ESCC and adjacent tissue. Our work will lay the foundation for developing and applying immune-targeted strategies for ESCC diagnosis and treatment.

Our study showed that ESCC was enriched in immune-suppressive cell populations, including Tregs, exhausted CD8 T, CD4 T and NK cells, M2 macrophages, and tDCs. All these immune-inhibitory cells may contribute to immune escape and promote tumor progression. Interestingly, we demonstrated that exhausted CD4, CD8 T cells, and NK cells were the major intra-tumoral proliferating immune cell compartments, although these cells were enriched in exhaustion genes. Pre-exhausted clusters CD8-C5-CCL5 and CD8-C6-STMN1 may serve as better targets for immunotherapies compared to the exhausted cluster (CD8-C7-TIGIT), as the latter are in a permanent and less reversible exhausted stage, making them more resistant to checkpoint inhibition due to their epigenetic changes[50,51]. Furthermore, we

found that tumor-infiltrating NK cells were not only commonly reduced in ESCC, but also expressed high levels of checkpoint molecules, including NKG2A and CD49d, suggesting an exhausted state. Anti-NKG2A and anti-CD49d have been reported to be checkpoint inhibitors that promote anti-tumor immunity[52,53]. Our results suggested alternative pathways to re-activate anti-tumor immunity in ESCC, such as the blockade of NKG2A and CD49d alone or in combination with anti-PD1/PD-L1, which may improve the immunotherapeutic response.

Clonal amplification of cells carrying identical TCR-seq across clusters was a strong evidence to support the connection among these clusters and indicated the transition of cell status. While most cells contained unique TCRs, clonal amplification was observed to varying degrees in different clusters. Indeed, we found sharing of TCR sequences in almost all CD4 and CD8 clusters, suggesting a broad differentiation after T cell priming. Notably, exhausted CD8 T cells (CD8-C7-TIGIT) carried a much higher percentage of shared clones with other CD8 clusters, especially with the pre-exhaustion clusters CD8-C5-CCL5 and CD8-C6-STMN1, which was consistent with the related status of these clusters and the multi-step exhaustion hypothesis. In contrast, CD8-C1-NKG7, the most cytotoxic CD8 T cell cluster, had significantly fewer clonal T cells in tumors. This result suggested the inhibition of cellular proliferation of cytotoxic CD8 T cells by TEM. This may contribute to the decreased numbers of tumor-infiltrating effector CD8 T cells and to the immune-suppressive microenvironment. A similar finding was recently reported in bladder cancer, in which cytotoxic CD8 T cells (FGFBP2+ cluster in their analysis) were more clonal in normal tissues than in tumors[54]. On the other hand, it is possible that the clonal amplification of esophageal cytotoxic cells was due to occasional exposure to non-tumor antigens in the esophagus, and that most CD8-C1-NKG7 cells in adjacent tissues express low levels of CD39, indicating that they are bystander CD8 T cells. Interestingly, we did not find a significant population of naïve CD8 T cells in our samples, and similar studies in other cancer types, e.g. melanoma and liver carcinoma, identified a limited number of naïve CD8 T cells[11,55]. It is possible that naïve CD8 T cells less frequently infiltrate into the esophagus or are activated by the local environment, making them less likely to be detected as an independent population. Additional studies are needed to validate this phenomenon and identify the mechanism.

The critical role of monocytes/macrophages in tumors has been described in liver, breast, and lung cancers using scRNA-seq. Typically, macrophage activation is classified into either a pro-inflammatory M1 state or an M2 state associated with the resolution of inflammation[56]. Our analyses revealed that monocytes/macrophages reside along a spectrum of monocyte, M1, and M2 states; both M1-associated and M2-associated genes were frequently co-expressed in the same cells. Zhang, et al. reported six macrophage clusters identified in hepatocellular carcinoma, and found that a macrophage cluster (M4-C1-THBS1 in their analysis) was enriched for signatures of MDSC, similar to

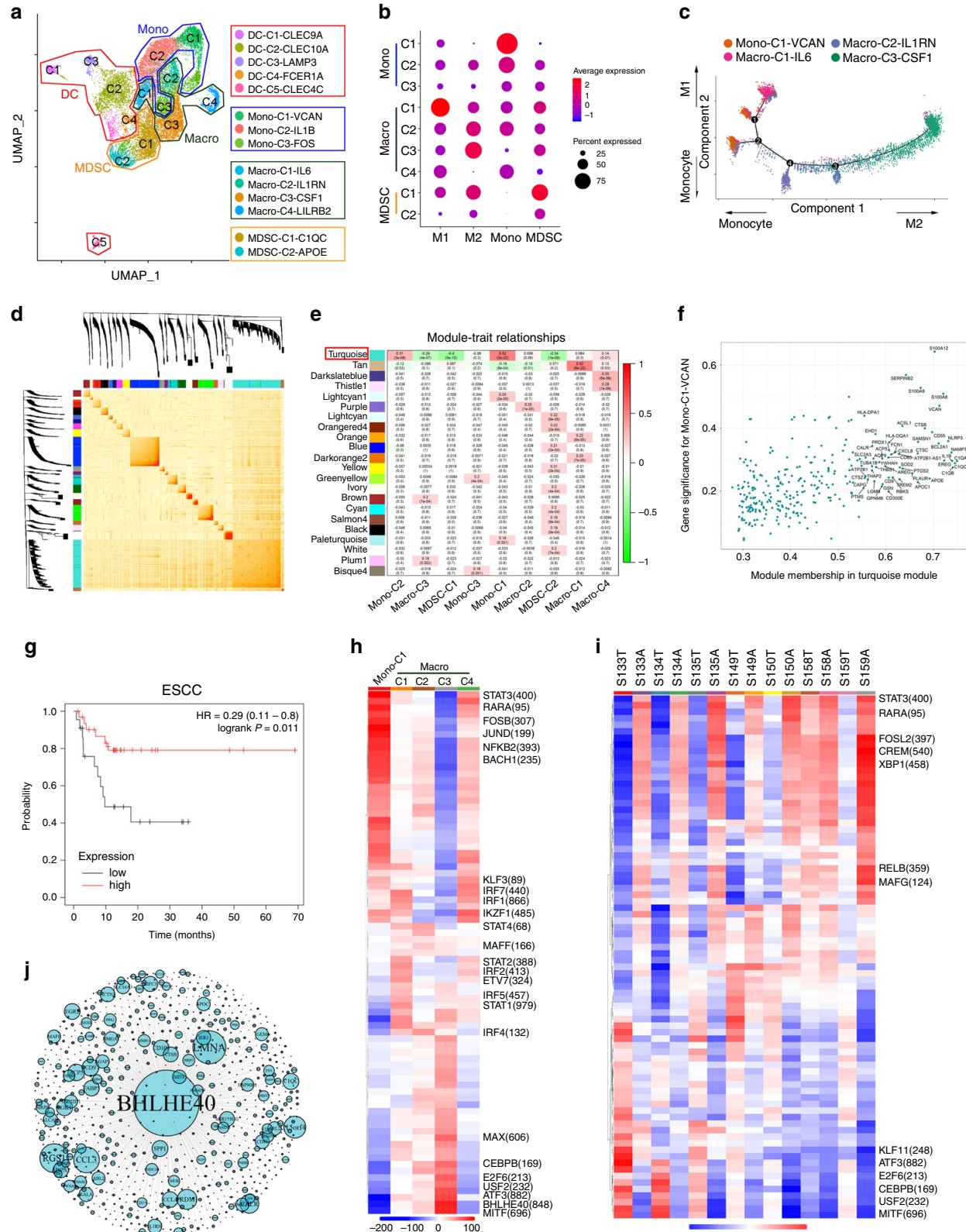

ours MDSC-C1-C1QC and MDSC-C2-APOE; they also found another cluster M4-C2-C1QA, co-existed with M1 and M2 signatures[20]. Lambrechts, et al. reported that macrophages in lung TME show rheostatic phenotypes and become M2 polarized[19]. Indeed, we found the M2 cluster Macro-C3-CSF1 was enriched in ESCC compared with adjacent tissue. The co-existence of M1 and M2 signatures indicated that TAMs were

more complex than the classical M1/M2 model, and this phenomenon was also found in breast cancer and liver carcinoma[12,18]. When using WGCNA to analyze gene correlation from monocytes/macrophages, we found a gene set that was positively related to monocytes, patients who expressed higher of this signature had significantly better prognosis in ESCC and other tumor types with similar pathology, suggesting that this

**Fig. 5 Detailed characterization of monocyte/macrophage cells. a** UMAP plot of 17,256 myeloid cells from 14 samples, showing the formation of the 14 main clusters, including nine for monocytes/macrophages (mono/Macro) and five for dendritic cells (DCs). Each dot corresponds to one single-cell, colored according to the cell cluster. **b** Dot plot of representative M1, M2, monocyte, and MDSC signatures in the monocytes/macrophage clusters. Z-score normalized log2 (count + 1). **c** The trajectory of Macro-C3-CSF1, Mono-C1-VCAN, Macro-C2-IL1RN, Macro-C1-IL6 state transition in a two-dimensional state-space inferred by Monocle. Each dot corresponds to one single-cell, colored according to its cluster label. Arrows show the increasing directions of certain cell properties. **d** A gene co-expression network of single cell from monocytes/macrophages was constructed by weighted correlation network analysis. The heatmap showed the topological overlap matrix among all genes used in the analysis. The darker color represented a higher overlap. The hierarchical clustering and module assignment of genes were shown along the left side and top. **e** Module–trait relationships in all monocytes/macrophages sub-cluster. Within each heatmap, red indicated a positive correlation and green indicated a negative correlation. The numbers in brackets were correlation P values. The red box indicated the Turquoise module was highlighted. **f** The Turquoise module membership and gene significance for Mono-C1-VCAN. The most correlated genes were labeled. **g** Kaplan–Meier overall survival curves of TCGA ESCC patients with the top 50 most correlated genes generated in **f**. P-values were calculated using the two-sided log-rank test. **h** Heatmap of the *t* values of AUC scores of expression regulation by transcription factors of the indicated clusters, as estimated using SCENIC. **i** Heatmap of the *t* values of AUC scores of expression regulation by transcription factors of the different samples, as estimated using SCENIC. A, Adjacent; T, Tumor. **j** Network analysis of the transcription factor BHLHE40 and its target genes through iGraph. The size of the circle represents the co-expression relationship score.

signature may serve as a prognostic biomarker in ESCC and other squamous cell carcinomas.

cDC2 is specifically involved in MHC class II-mediated antigen presentation and the activation and expansion of CD4 T cells. cDC1 is also necessary for anti-tumor immunity. Recently, some scRNA-seq papers have reported DCs in the TME[12,40,57]. We found that most of them were consistent with our data. Zilionise et al. reported that human DCs contain four distinct subsets, including hDC1, hDC2, hDC3 and pDC, hDC1, hDC2, and hDC3 were consistent with our cDC1 (DC-C1-CLEC9A), cDC2 (DC-C2-CLEC10A), and Lamp3+DC (DC-C3-LAMP3) subsets, respectively. However, the monocyte-derived DC (DC-C4-FCER1A) was absent in that paper. These differences may reflect a variation in DC states between different tumor tissues and/or the setting of analysis. Interestingly, we found that LAMP3 + DC had multiple functions, such as activation activity, migration activity, and tolerogenic ability, and this subset was also found and reported in lung and liver cancers, suggesting the conserved myeloid cells exist in many tumors.

Our work further demonstrated the interaction between Tregs and macrophages which may contribute to the immunosuppressive microenvironment in ESCC. Our data suggested that IL1R2 expressed on Tregs may enhance Treg function by blocking IL1β-dependent effector T cell activation. IL1R2, which serves as the IL1β decoy receptor and binds to IL1β can block follicular cell activation[58]. It has been reported that IL1R2 expressed on activated tumor Tregs and is correlated with poor prognosis in lung adenocarcinoma[59,60]. Our study also demonstrated that Tregs may modulate macrophage function through HLA-A, B, C, and LILRB1 interaction. LILRB1 expression was upregulated on TAMs, disruption of either MHC class I or LILRB1 potentiated phagocytosis of tumor cells both in vitro and in vivo. Investigating the mechanisms underlying this immuno-suppressive MHC class I-LILRB1 signaling axis in TAMs will be useful in developing therapies to restore macrophage function[46,61], and blocking this pathway may promote antitumor immunity in ESCC.

In summary, our transcriptional map of immune cells from ESCC and adjacent tissue provided a framework for understanding the immune status and revealed the dynamic nature of immune cells in the ESCC setting. In addition, we illustrated the immune-suppressive state of ESCC from many aspects, all of which are potential novel targets for developing immunotherapies in ESCC and other cancers.

## Methods
**Sample collection and preparation.** Seven patients who were pathologically diagnosed with ESCC were enrolled in this study for single cell RNA-seq analysis.

None of the patients had been treated with chemotherapy, radiation, or any other anti-tumor medicines prior to tumor resection. The clinical information of these patients was summarized in Supplementary Table 1. Paired, freshly excised ESCC tumors and adjacent esophageal tissues were obtained immediately after surgical removal. The adjacent tissues were at least 5 cm from the tumor tissues. Clinical samples were collected from the Xinhua and Ruijin Hospital, Shanghai Jiaotong University School of Medicine. Prior to participation, written informed consent was obtained from all subjects. All studies were performed in accordance with the Declaration of Helsinki. The study was approved by the Research Ethics Board of the Xinhua and Ruijin Hospitals, Shanghai Jiao Tong University School of Medicine.

Tumor and adjacent tissues were isolated by mincing the freshly obtained surgical specimens into 1-mm cubic pieces, followed by enzymatic digestion using 0.1% collagenase IV, 0.002% DNAse I, and 0.01% hyaluronidase, and were incubated on a rocker for 20–40 min at 37 °C. The digested tissues were then passed through a 40 μm cell strainer and washed twice with PBS prior to surface staining. Immune cells were stained at $1 \times 10^6$ cells per ml with antibodies (CD45-APC, CD235-FITC) for 30 min at 4 °C, and then washed and suspended in 200 μL FACS buffer. PI was added 5 min before flow cytometry sorting for dead cell discrimination. CD45+ CD235−PI−cells were sorted with FACS Aria II Cell Sorter (BD Biosciences); post-sort purity was routinely >95% for the sorted populations. The CD45+CD235− PI− cell gating strategy is shown in Supplementary Fig. 12a.

**RNA-Seq library preparation for 10× Genomics single-cell 5′ and VDJ sequencing.** FACS-sorted CD45+ cells were suspended at a $1 \times 10^6$ cells/ml concentration in FACS buffer and the viability was higher than 85%. Single-cell library preparation was carried out according to the manual of the Chromium Single-cell V (D) J Reagent Kits. Briefly, cell suspensions were first loaded on the Single-cell A Chip for gel bead-in-emulsion (GEM) generation and barcoding, aiming for a recovery of 10,000 cells per sample. Next, reverse transcription, RT cleanup cDNA amplification and quantification were performed to build a 5′ gene library. In addition, a human T Cell V (D) J Enrichment Kit was used to isolate and enrich for the V (D) J sequence before TCR library construction. Finally, libraries were sequenced on the NovaSeq 6000 system with NovaSeq 6000 S4 Reagent Kit (300 cycles).

**scRNA-seq data analysis.** The 10× Genomics Cell Ranger (3.0.1 version) pipeline was used to demultiplex raw files into FASTQ files, extract barcodes and UMI, filter, and map reads to the GRCh38 reference genome, and generate a matrix containing normalized gene counts versus cells per sample. This output was then imported into the Seurat (v3) R toolkit for quality control and downstream analysis. All functions were run with default parameters, unless otherwise specified. Low-quality cells (<400 genes/cell and >10% mitochondrial genes) were excluded. As a result, 80,787 cells with a median of 1170 detected genes per cell were included in downstream analyses. To remove the batch effect, the datasets collected from different samples were integrated using Seurat v3 with default parameters.

**Dimensionality reduction, clustering, and annotation.** We then identified a subset of genes that exhibit high cell-to-cell variation in the dataset, which helped to represent the biological signal in downstream analyses. The Seurat function 'FindVariableFeatures' was applied to identify the highly variable genes (HVGs). The top 2000 HVGs were used for data integration. The data were scaled using 'ScaleData' and the first 20 principle components were adopted for auto-clustering analyses using 'FindNeighbors' and 'FindClusters' functions. For all 80,787 cells, we identified clusters setting the resolution parameter as 1.5, and the clustering results were visualized with the UMAP scatter plot. The marker genes of each cell cluster

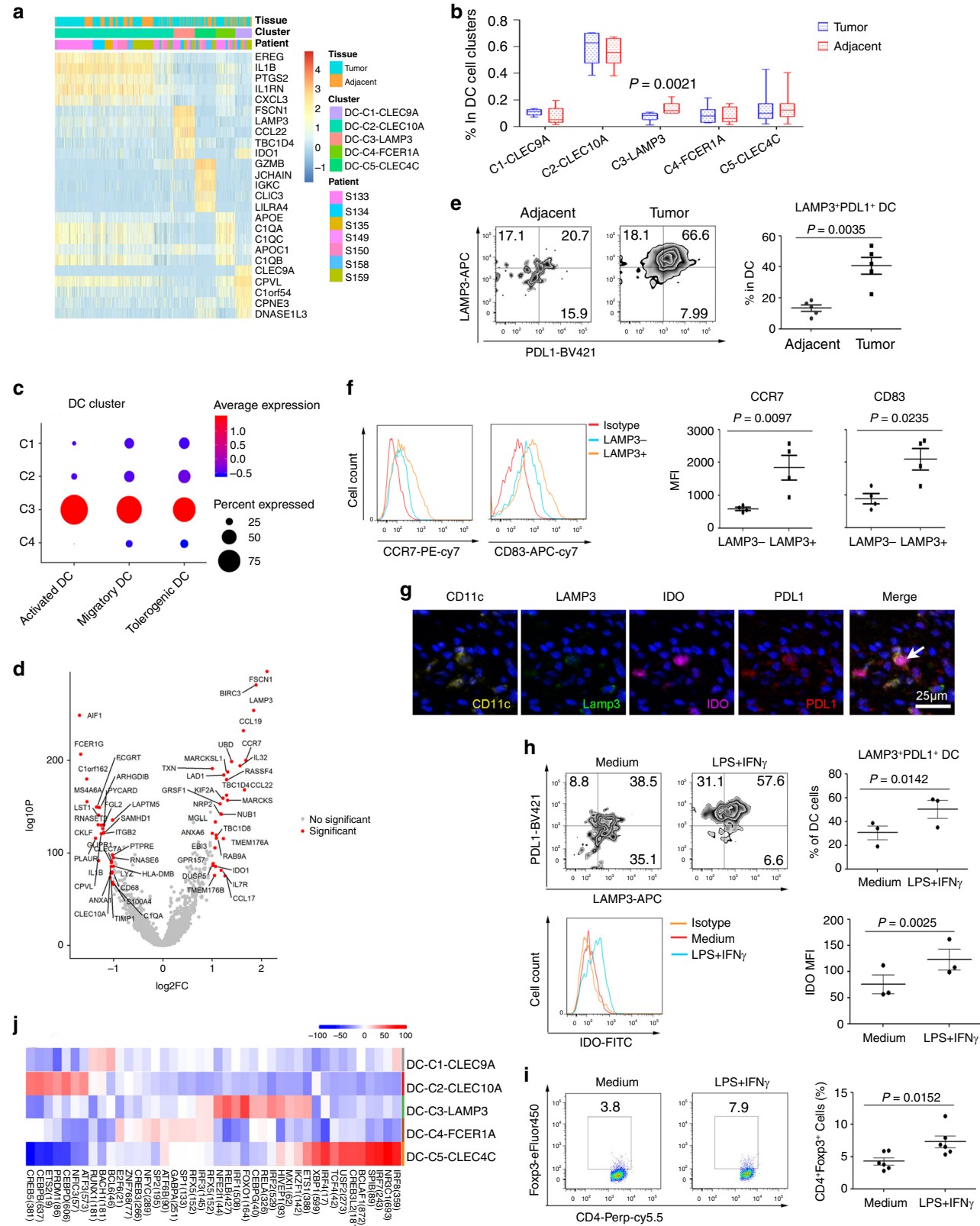

were identified using the ROC analysis function provided by the Seurat 'FindAll-Markers' function for the top genes with the largest AUC (area under curve). We used GSEA (http://software.broadinstitute.org/gsea/msigdb/annotate.jsp) to perform biological process enrichment analysis on the differentially expressed genes in each cluster or subset. The whole dataset was then categorized into NK cells, T cells, monocytes/macrophage cells, B cells, mDCs, pDCs, plasma cells, mast cells, and other cells (including fibroblast and basal cells) according to the known markers: *KLRC1*, *KLRD1* (NK cells), *CD3G*, *CD3D*, *CD3E*, *CD2* (T cells), *CD14*,

*VCAN*, *FCGR2A*, *CSF1R* (monocyte/macrophages), *CD19*, *CD79A* (B cells), *CD1C*, *FCER1A* (mDC), *CLEC4C* (pDC), *SLAMF7*, *IGKC* (plasma cells), *TPSB2*, *CPA3* (mast cells), *KRT19*, *IGFBP4*, *CTSB* (basal cells/fibroblasts). Clusters were also confirmed by identifying significantly highly expressed marker genes in each cluster and then comparing them with the known cell-type-specific marker genes. The 44,634 NK-T cells were further categorized into 17 clusters with a resolution parameter of 1. The 17,256 myeloid cells were further categorized into 14 clusters with a resolution parameter of 0.4.

**Fig. 6 Detailed characterization of DC. a** Heatmap of five DC cluster marker genes. The top bar indicated the origins, and the middle bar indicated the clusters, and the lower bar indicated the different patients, the color intensity indicated the expression level. The expression was measured as Z-score normalized log2 (count + 1). **b** Box plots of each dendritic cluster between the tumor and the adjacent tissues (n = 7). Each box represented the interquartile range (IQR, the range between the 25th and 75th percentiles) with the mid-point of the data; whiskers indicated the upper and lower values within 1.5 times the IQR. P value was calculated by two-tailed paired Student's t-test. **c** Dot plot representative of the activation, migration, and tolerogenic signatures in dendritic clusters. The expression was measured as Z-score normalized log2 (count + 1). **d** Volcano plot showing the differentially expressed genes (LAMP3+ DC versus other DC clusters), each red dot denoted an individual gene with an adjusted P value < 0.01 and fold change > =2, two-sided Wilcoxon test. **e** FACS analysis of LAMP3 and PDL1 expression in DC cells from adjacent and ESCC tumor tissues (n = 5). Data were presented as mean ± SEM; P value was calculated by paired two-tailed t-test. **f** CCR7 and CD83 expressions in LAMP3+ and LAMP3− DC cells from adjacent and ESCC tumor tissues were measured by FACS (n = 4). Data were presented as the mean ± SEM. P value was calculated by paired two-tailed t-test. **g** Detection of CD11C +LAMP3+PDL1+IDO+ DCs in ESCC tissue by multi-color IHC staining. Representative data from three patients were shown. **h** DC cells were stimulated with or without LPS and IFNγ for 24 h, and the expression of PDL1, LAMP3, and IDO was analyzed. One of the three similar experiments was represented; data were presented as the mean ± SEM, P value was calculated by two-tailed Student's t-test. **i** DC cells were stimulated with or without LPS and IFNγ or not for 24 h and cultured with CD4+CD45RA+ naïve T cells for 4 days, and FOXP3 expression were measured. One of the six similar experiments was represented, data were presented as the mean ± SEM, and P value was calculated by two-tailed Student's t-test. **j** Heat-map of the t values of AUC scores of expression regulation by transcription factors of the indicated clusters, as estimated using SCENIC.

**Definition of exhaustion, naivete, cytotoxicity, and other scores**. We considered published signature gene lists for cytotoxicity, exhaustion, Treg, M1, M2, monocyte, and MDSC (Supplementary Data 3), and the DC cell activation, migration and tolerogenic scores have been previously described[62] and are provided in Supplementary Data 5. We used the Z-scores of the average expression of four well-defined naïve markers (CCR7, TCF7, LEF1, and SELL) to define the naïve gene score for both CD8 and CD4 T cells.

**Development of trajectory inference**. We applied Monocle (version 2) and Slingshot using default parameters on the identified monocyte and macrophage cells to determine the potential lineage differentiation. First, the top 1000 HVGs were selected using the function 'FindVariableFeatures' in Seurat v3. Then the 'estimateSizeFactors', 'estimateDispersions', and 'dispersionTable' functions with default parameters were used to build statistical models to characterize the data. The genes with mean expression >0.1 were retained for subsequent analyses. The dimensionality reduction was performed by the 'reduceDimension' function using the DDRTree method.

**SCENIC analysis**. The SCENIC analysis was run as described[38], using the pyscenic (version 0.9.19) and hg19-500bp-upstream-10species databases for RcisTarget, GRNboost, and AUCell. The input matrix was the normalized expression matrix that was from Seurat.

**Cell–cell interaction**. Cell–cell interaction analysis was analyzed by two algorithms, one has been described previously[43], and the other was scTHI (https://github.com/miccec/scTHI). We determined the existence of the potential interactions between two cell types mediated by evaluating the expression levels of annotated ligands and receptors between the two cell types. A ligand–receptor score for each sample was calculated by multiplying the average expression of a ligand from one cell type and the average expression of a receptor from the other cell type. The interactions between the two cell types via the ligand–receptor pair were identified when the scores of the samples from the seven patients were significantly larger than zero (Wilcox test P < 0.05).

**TCR data analysis**. A 10× Genomics Cell Ranger pipeline was used to identify clonotypes by alignment and annotation with the default settings, according to the manufacturer's recommendations. TCR reads were aligned to the GRCh38 reference genome, and consensus TCR annotation was performed using Cell Ranger VDJ. Only in-frame TCR alpha–beta pairs were considered to define the dominant TCR of a single-cell. Each unique dominant alpha–beta pair was defined as a clonotype. For two cells to be assigned to the same clonotype, both alpha and beta sequences had to be shared. If one clonotype was present in at least two cells, cells harboring this clonotype would be considered clonal, and the number of cells with such dominant alpha–beta pair indicated the degree of clonality of the clonotype.

**Survival analysis**. An online website (http://kmplot.com/analysis/index.php?p=service)[63] was used to analyze the relationship between BHLHE40, and the 50-gene signature and ESCC survival.

**Flow cytometric analysis**. For cell surface marker staining, antibody cocktails were used (CD45, PI, CD3,CD4, CD8, CD19, CD138, CD25, CD127, CCR7, CD83, PD1, CD68, CD163, CD206, CD11C, CD14, CD56, NKG2A, CD66B, CD15, CD11B, CD117, HLA-DR, PDL1, CD11C, CD121b, LILRB1) For intracellular cytokine staining, cells were stimulated with a cell stimulation cocktail plus protein transport inhibitors (eBioscience) for 5 h. Then, the cells were fixed and

permeabilized with Cytofix/Cytoperm buffer, and intracellular cytokines were stained with antibodies against FOXP3, IFNγ, TNFα, IDO1, LAMP3, and isotype control according to the manufacturer's instructions. The antibody dilutions are in accordance with the instructions. Flow cytometric analysis was performed with a FACS Canto II instrument (BD Bioscience) and FlowJo software (TreeStar). Figure 3c, f, i gating strategies were shown in Supplementary Fig. 12b–d. Figure 6e gating strategies were shown in Supplementary Fig. 12e. Figure 6f gating strategies were shown in Supplementary Fig. 12f. Figure 6h gating strategy was shown in Supplementary Fig. 12g. Figure 6i gating strategy was shown in Supplementary Fig. 12h. Figure 7e, f gating strategies were shown in Supplementary Fig. 12i. Supplementary Figs. 7i and 9e gating strategies were shown in Supplementary Fig. 12j. Supplementary Fig. 11a, f gating strategies were shown in Supplementary Fig. 12k. Figure 7k, l gating strategies were shown in Supplementary Fig. 12l.

**In vitro stimulation of DC subsets**. Fresh blood was obtained from 30 healthy volunteers, (age range from 34 to 62, median age is 46). Peripheral blood mononuclear cells (PBMCs) were isolated from heparinized fresh blood by standard density gradient centrifugation with Ficoll-Paque Plus (GE Healthcare). Subsequently, Pan-DCs were obtained by negative selection using a Human Pan-DC pre-enrichment kit (Stem Cell Technologies). DCs were cultured in CellGenix GMP DC medium (CellGenix), and stimulated with LPS-EB Ultrapure (Sigma-Aldrich) and INFγ (Peprtotech); both at 100 ng/ml or not for 24 h, and then co-cultured with CD4+CD45RA+ naïve T cells in the anti-CD3 (eBioscience) 1 µg/ml for 4 days, and FOXP3 expression was detected by flow cytometry.

**Macrophage differentiation**. To obtain macrophage cells from healthy PBMC, the cells of each subject were cultured by using RPMI 1640 (Gibco) medium that included 10% fetal bovine serum (FBS) (Gibco), 100 U/ml penicillin, 100 µg/ml streptomycin (Gibco) and 1% 2 mM L-glutamate (Gibco) at 37 °C in a %5 $CO_2$ incubator for 4 h. After the culture period, non-adherent cells were washed with phosphate-buffered saline (PBS) and cultured in different culture plates by using RPMI 1640 complete medium and 10 ng/ml human IL-2 (Peprotech) until to co-culture experiments. Adherent cells were detached by trypsin-EDTA (Gibco) and divided into $1 \times 10^5$ cells per well in two 12-well plates. Cells were cultured using RPMI 1640 complete medium and human 25 ng/ml macrophage colony-stimulating factor (M-CSF) (Peprotech) at 37 °C in a 5% $CO_2$ incubator for 4 days.

**IL1R2 antibody blocking**. FACS sorted CD4+CD25− Teff cells were stained with CFSE (Sigma-Aldrich) at 2 µM for 8 min at room temperature. A total of $3 \times 10^4$ Teff cells were stimulated with or without IL-1β at 20 ng/ml, then co-cultured with or without FACS-sorted CD4+CD25hiCD127lo/- Treg cells at $3 \times 10^4$ in the presence of anti-CD3 1 µg/ml, anti-CD28 at 0.5 µg/ml (eBioscience), stimulated with anti-IL1R2 or control IgG (R&D) at 20 µg/ml, and then cultured for an addition 72 h. The expression of IFNγ and the division of Teff cells were analyzed by flow cytometry.

**LILRB1 antibody blocking**. Macrophages ($3 \times 10^4$) were co-cultureed with FACS sorted CD4+CD25hiCD127lo/- Treg cells ($3 \times 10^4$) in the presence of anti-CD3 1 µg/ml, stimulated with anti-LILRB1 or IgG control (R&D) at 10 µg/ml, and cultured for an additional 48 h. The expression of CD163, CD206, PDL1, and TNFα in macrophage cells was analyzed by flow cytometry.

**Multi-color IHC**. Human tissue specimens were provided by Xinhua and Ruijin Hospital, Shanghai Jiaotong University School of Medicine under an approved Institutional Review Board protocol. The specimens were collected within 30 min

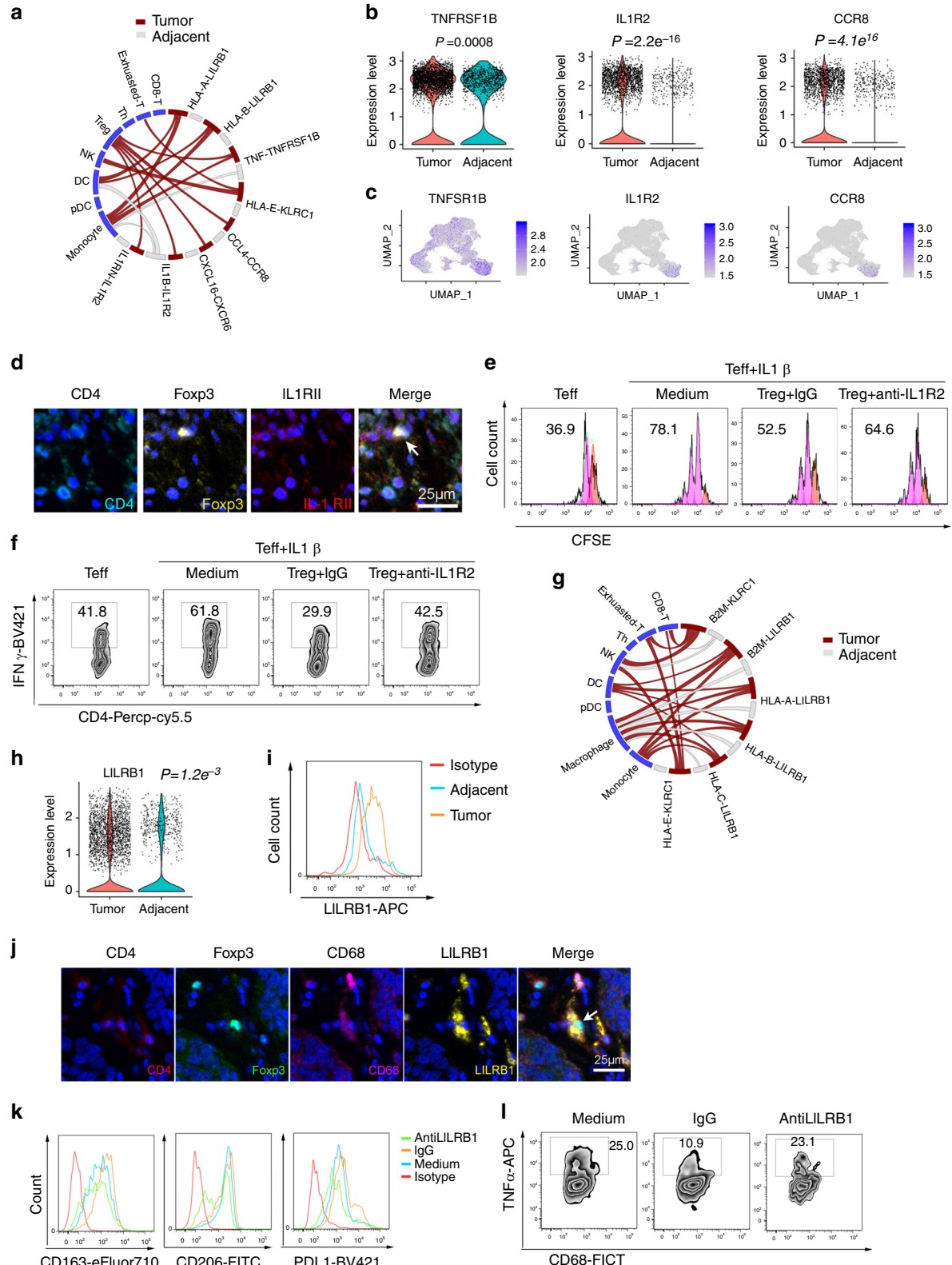

after tumor resection and fixed in formalin for 48 h. Dehydration and embedding in paraffin were performed following routine methods. These paraffin blocks were cut into 4 mm slides and fixed to glass slides. Then the paraffin sections were placed in the 70 °C paraffin oven for 1 h before deparaffinization in xylene and then rehydrated in 100%, 90%, 70% alcohol successively. Antigen was retrieved by immersion in boiling EDTA buffer (pH 9.0) for 15 min. Following a preincubation with Antibody Diluent/Block (PerkinElmer) to block nonspecific sites for 15 min, the sections were incubated with primary antibodies in a humidified chamber at 4 °C overnight. After the sections were washed with PBS twice for 5 min, they were incubated with Opal Polymer HRP Ms+Rb (PerkinElmer) for 10 min at 37 °C. The antigenic-binding sites were visualized using the Opal Seven-Color IHC Kit (PerkinElmer) according to the manufacturer's protocol. Images were captured and analysis was performed with PerkinElmer Vectra 3.0.5 (PerkinElmer) and informs (PerkinElmer), respectively.

The primary antibodies, IHC metrics and fluorophores used in the validation of tolerogenic LAMP3+DC, rabbit anti-human CD11C (Abcam, 1:200) (Oapl690), rabbit anti-human LAMP3 (Abcam, 1:100) (Opal620), rabbit anti-human IDO1 (Abcam, 1:100) (Opal520), and rabbit anti-human PDL1 (Abcam, 1:100)

**Fig. 7 Ligand–receptor-based interaction between immune cells in ESCC. a** Circos plot showing the predicted average strength in seven patients of the indicated interaction between macrophage-provided ligands and receptors in other cell types within tumor or adjacent tissues. **b** Violin plot showing the average expression value of a given gene in CD4-C6-FOXP3 from the tumor and adjacent tissues. The expression was measured as the log2 (count + 1), two-tailed Wilcoxon test. **c** Expression levels of relative marker genes across NK-T cells illustrated in UMAP plots, the expression was measured as the log2 (count + 1). **d** Detection of CD4, Foxp3, and IL1RII in ESCC tissue by multi-color IHC staining. Representative data from three patients was shown. **e**, **f** CD4+ CD25− (Teff) cells were stimulated with IL1β, Treg cells were co-cultured with Teff cells in the presence of anti-IL1R2 or IgG antibodies, the percentage of Teff dividing cells ($n = 6$) **e** and the IFNγ expression in Teff cells ($n = 4$) **f** was measured by FACS after 72 h culture. **g** Circos plot showing the predicted interaction strength of the indicated L–R pairs between tumor and adjacent tissues as Treg-provided ligands and interactions with the indicated cell types averaged across seven patients. **h** Violin plot showing the average LILRB1 expression in macrophages from tumor and adjacent tissues. The expression was measured as the log2 (count + 1), two-tailed Wilcoxon test. **i** FACS analysis of LILRB1 expression in CD68+ macrophage cells from adjacent and ESCC tumor tissues ($n = 3$). **j** Detection of CD4, Foxp3, CD68, and LILRB1 in ESCC tissue by multi-color IHC staining. Representative data from three patients was shown. **k**, **l** Macrophages were co-cultured with or without Treg cells, and flow cytometry measured the expression of CD163, CD206, and PDL1. Representative data from four patients was shown **k** and TNFα expression. Representative data from three patients was shown ($n = 3$) **l** in CD68 macrophage cells in the presence of anti-LILRB1 or control IgG antibody.

(Oapl570). The validation of the potential physical interaction (co-localization) between macrophages and Treg cells were: rabbit anti-human CD4 (Abcam, 1:500) (Oapl690), mouse anti-human FOXP3 (Abcam, 1:100) (Oapl650), mouse anti-human CD68 (Abcam, 1:100) (Opal620), and rabbit anti-human LILRB1 (Abcam, 1:500) (Opal520). IHC was performed using a fully automated IHC instrument BOND MAX (LEICA). The primary antibodies used were anti-human CD3 (LEICA, 1:200), anti-human CD20 (LEICA, 1:300), anti-human MPO (Long island, 1:100), anti-human CD56 (Long island, 1:400), and anti-human CD68 (Long island, 1:100). All the antibodies information is included in Supplementary Table 2.

**Weighted gene co-expression network analysis (WGCNA)**. To construct the gene co-expression network, the R package WGCNA was implemented and a signed network was constructed using genes that were selected as noise robust by OGFSC using default parameters[64]. To group genes with coherent expression profiles into modules, average linkage hierarchical clustering was applied, using the topological overlap measure as the dissimilarity. The modular gene centrality, defined as the sum of within-cluster connectivity measures, was used to rank modular genes for hubness within each gene module.

**Statistical analysis**. The statistical methods used for each analysis are described in the above "Methods" sections and in the figure legends.

**Reporting summary**. Further information on research design is available in the Nature Research Reporting Summary linked to this article.

## Data availability

The Single cell RNA-sequencing data used in this study are available in the Gene Expression Omnibus (GEO) database under accession code GSE145370. The remaining data are available within the Article, Supplementary Information or available from the authors upon request.

## Code availability

The codes generated during this study are available at the Github repository (http://github.com/xinhua-lab/sc-hESCC).

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

## Acknowledgements

We thank Prof. Bin Li for his valuable scientific advice and discussion. This work was supported by grants from the National Natural Science Foundation of China [81571525 and 81873863 to Y.Z., 81672363 to L.S., 81871882 to H.L.], the Shanghai Municipal Education Commission—Gaofeng Clinical Medicine Grant Support [20161315 to Y.Z.], and the Shanghai Pujiang Talent Plan [17PJD027 to Y.Z.], the Key Specialty Development Program of Xin Hua Hospital and Shanghai Municipal Health Commission (to L.S.), the Clinical Research Plan of SHDC (Project No. 16CR3057A, to L.S.), the Medicine and Engineering Cross Research Foundation of Shanghai Jiao Tong University (Project No. YG2017ZD02 to L.S.), the Start-up Fund and a New Investigator Award (both to J.C.) provided by Rutgers Cancer Institute of New Jersey (State of NJ appropriation and National Institutes of Health grant P30CA072720), the Rutgers School of Public Health Pilot Grant and the New Jersey Alliance for Clinical and Translational Science Mini-methods Grant to W.V.L. The single cell RNA-seq was supported by Genergy Biotechnology of Shanghai. The data analysis was supported by GorgeosAI (Gu Ai) Biotechnology and the Program of Innovative Research Team of High-level Local Universities in Shanghai.

## Author contributions

L.S., J.C., H.L., and Y.Z. designed experiments. Y.Z., Z.C., L.H., Y.H., B.Z., R.H., H.X., and S.C. performed the experiments. Y.Z., Z.C., X.Z., W.V.L., J.H., S.B., Y.M., G.X., J.Y. analyzed sequencing data. L.S., J.C., Y.Z., A.B., and W.V.L. wrote the manuscript, with all authors contributing to writing and providing feedback.

## Competing interests

The authors declare no competing interests.
