## [Peer Review File · Nature Communications]

Reviewers' Comments:

Reviewer #1:

Remarks to the Author:

In this manuscript by Zheng et al, the authors applied 10x single cell technology to obtain RNAseq profile from 7 patients with esophageal SCCs, and additional TCR-sequencing for the T-cell component from these samples. They focused their attention on flow-sorted CD45+ cells only (ie immune compartment), from tumor and adjacent normal (more than 5 cm apart) tissue, and therefore managed to extract RNAseq data from a total of 80K+ single cells, surprisingly in some circumstances even getting 9000 cells per sample. They subsequently delve into the different subsets, to derive insights into the immune microenvironment that exists in these cancers which naturally have a poor outcome with little in the way of targeted therapeutic options. In doing, they put together immune micro-environment of ESCC which can be subsequently used to compare either with other tumors of the digestive tract OR with similar histologies (ie SCCs) from other subsites. From a general perspective, this is a well-executed descriptive atlas with an opportunity to shed light on specific details of the immune milieu in ESCC. However, in studies like these, there is a need to up front decide whether the data is purely descriptive or with functional intent. Supporting experiments can therefore be focused to support the former or latter intent. The main issue in this study that cuts across the entire paper is that it is meant to be descriptive, but there is little to no attempt to validate these findings with supportive data, and that is a critical omission. In contrast, the authors present a single functional experiment (which was not the objective of this paper to begin with), which confounds, rather than illuminates (see below). Secondly, there is also a need to link some of these data in a logical manner in relation to known aspects of tumor immunology, and this is also missing in many sections (see below), and therefore needs to be addressed. Finally, data about cell-cell interactions (last figure) are speculative at best, and investing such a large proportion of the paper to this, without any functional or supportive data invites criticisms. Nonetheless, I feel that if many of these issues can be resolved, this data is still an important reference that warrants its publication. Below I have listed down a number of issues that I feel are important revisions required to be addressed.

1. One of the major claims in Figure 1 and 2 are the proportions of the different immune cell lineages, comparing tumor versus normal and across different patients. While the data for these differences are interesting, it needs to be backed up by other methodologies or orthogonal data- flow cytometry, IHC, bulk sequencing of unprocessed samples. The main issue is that processing time and methodology can greatly alter specific lineages, and these numbers may be skewed between and across experiments. Neutrophils and macrophages are especially vulnerable in these situations, and statements about proportions of these need to be validated before being stated as fact. Even across all the normal samples, there is significant variation using scRNAseq. The data shown here could even be validated against published bulk RNAseq data comparing tumor vs normal.
2. There is insufficient data about tumors-normal pairs, especially given the variation seen across adjacent normal...were there anything specific about those with higher T cells compared to the rest? Were they closer to the tumor? I would like to see a scaled diagram or cartoon showing the location of each tumor and normal pair.
3. There is little to no data about the associated tumor in each case as well? Do you know the tumor fractions of the region used for these above analyses? Were they at the middle or periphery of the cancers? There is no profiling data of the tumor cells presented.
4. I do not understand the part of data that states "We found inter-patient variation in biologic signatures, including hypoxia, epithelial-mesenchymal transition, inflammation response, and TNFA-via NFkB pathways", when these were based on profiling immune cells...would you expect EMT or hypoxia in these cells?
5. Data on the T cell compartment was described in detail but I felt that the links and connections made within this dataset, and what is known in the literature very scanty. For example, demonstrating that cytotoxic cells were more likely present at the adjacent normal regions is an important point, but was not elaborated further in the context of other cancers which describe these as being within the tumor as bystanders. Similarly the clusters of exhausted cells were well described, but there was no

proper connection made between these. The use of monocle in this context is also somewhat inappropriate. Monocle is used to demonstrate evolution of a cell type or cell state transitions and there is no evidence shown that the CD8 cells are derived from a common origin. In fact the authors are unable to demonstrate a convincing naïve population in their tumors at all!

6. Monocle data for the CD4 populations is even more illogical, as these are not subpopulations that transition from one to another in the tumor. Are you suggesting that Tregs and Th cells are matured in the tumor? If not, it does not make sense to align this in the same trajectory. There is little biological basis to use what is a powerful tool.

7. I found that the functional experiment introduced in figure 3e a bit random, not to mention the fact that the most important conclusion of increased TNF-alpha after CTLA4 blockade, was unconvincing and barely reached statistical significance. Moreover, the lack of activation of IFNG and GZMB is also not supportive of their conclusion that this blockade targets Tregs, which then affects CD8 cells. There are so many missing steps to draw this conclusion presented here, and this can either be addressed or omitted completely.

8. There are lineage connections within the CD4 and CD8 populations that can be drawn but these have not been highlighted or presented here at all.

9. TCR-sequencing was an important addition to this dataset, and could be used to address point 8. This was done somewhat for the tregs when the authors conclude that "This suggests that the high frequency of Treg cells were derived from naïve CD4 T cells and expanded locally in the ESCC microenvironment."

10. What proportion of T cell did they have TCR data for? It is difficult to make comments about TCR clones and clustering, and in fact this may be a biased analyses of better quality cells rather than distribution of TCR-clonality. It is intriguing that cytotoxic bystander cells in the adjacent normal tissue were more clonal compared to cells within the tumor itself, can the authors speculate why this might be so? Or if this is a phenomenon relevant to a contaminated site like the esophagus. Unfortunately the lack of attempt to link this data makes this section extremely descriptive with little or no major conclusions drawn.

11. Analyses of the myeloid component was probably the highlight of this paper and the data most intriguing. Again, I would have liked to have seen supportive flow cytometry data, especially to address the missing neutrophil populations herein. Trajectory analyses was also important and appropriate (for once) showing directionality from immature monocytes to M1 or M2, however I was unsure if Figure 5c was showing all the myeloid cells or only cluster 1,5,6,8? And if so, why and how were these chosen? Again Scenic analyses was appropriate here, but the authors did not link these with downstream genes through network analyses nor associate this data with previous publications. Data on DCs were also descriptive.

12. The last section of the paper was probably the most speculative. Many of these associations are known and the flaw of the methodology used only analyzes expression levels of ligand receptor pairs. I would have liked to have seen specific analyses of downstream and pathway analyses of interesting pathways that the authors were trying to highlight, if not this is just a long meandering speculative section.

13. Even the survival analyses based on cell-cell interaction was rather mundane. They just picked a macrophage M2 signature and showed its expression levels were prognostic in ESCC (lowish p value) and the in a bunch of other cancers which have no similarity to ESCC. The least I expected here was to use a composite signature that took into account co-expression models to extract this data and apply this across other SCCs where the conclusion actually makes biological sense!

14. My last point relates to editing. There are numerous spelling and grammatical errors that need to be addressed throughout the manuscript and figures.

Reviewer #2:

Remarks to the Author:

Comments to the author:

Zheng et al. are the first to comprehensively analyze the tumor microenvironment of esophageal squamous cell carcinoma (ESCC) based on single-cell RNA-seq and coupled T cell receptor (TCR)-seq of 80,787 immune cells derived from seven surgically removed ESCC tumors and their matched adjacent normal tissues. The authors observe a correlation between immunosuppression and the failure of immunosurveillance in ESCC, and eventually focus on the interaction of macrophages and NK cells which may contribute to the immunosuppressive status. Moreover, they identified a six-gene signature that is associated with poor survival in ESCC and other cancers. Overall, the group shows findings that would be of interest to scientists in the field of ESCC due to increasing interests to profile the immune status of ESCC in order to evaluate the application of current checkpoint blockade therapies and to develop innovative immunotherapies. However, with regard to the broader community and the wider scope of single-cell research in immunotherapy, the work does not influence thinking in the field. Most of the analyses are descriptive and sometimes lack a clear rationale.

Major comments:

1. Overall, the paper lacks a bit novelty and the data have not been interpreted sufficiently. The authors are the first to profile the tumor microenvironment in ESCC, but very similar approaches and analyses as seen across other cancer types. Some conclusions are not well underpinned, unclear or sometimes missing, e.g.:
 - Immunosuppression has already been indicated to be responsible for failure of immunosurveillance before (1)
 - The proposed six-gene signature which they found to be associated with poor survival in ESCC contains genes which have already been shown to be predictive in ESCC and other cancer types, such as IL-10 and CD274 (PD-L1) (2,3)
 - The techniques mentioned are novel in ESCC, however they could have used additional algorithms to further dig into the single cell data, e.g. to study cell interactions (NicheNet, CellPhoneDB...), copy number variation in cancer cells (InferCNV)...
2. It would be interesting if the authors also investigate malignant cells which they didn't cluster separately and annotated as "other". Gaining knowledge regarding the interactions between cancer cells and immune cells is fundamental to profile response to immune checkpoint therapy.
3. Their signatures for cytotoxicity, exhaustion, and regulatory T cells are based on the top 50 genes highly correlating with the expression of one single gene, i.e., FGF2, LAG3, and FOXP3, respectively (Fig. 2c). In other words, they conclude the functional state of cells by the expression of one anchor gene, which is not sufficient.
4. Many of their clusters seem to be driven by cell cycle genes. Why didn't they regress for this? Same accounts for IFN- and stress-related markers.
5. The findings related to trajectory inference are highly dependent on the specific tool used to perform the analysis. It is strongly suggested to verify their findings for trajectory inference of T cells and monocytes/macrophages (Fig. 2, 5) using additional publicly available tools such as SCORPIUS, Slingshot, etc.
6. The authors point out the difference between a "pre-exhaustion" and "exhaustion" state. This resembles the concept of "progenitor" vs. "terminally exhausted" T cells. It is recommended to check the markers that are already published for these specific states before concluding too quickly that the cells are "pre-exhausted", namely solely based on a more moderate expression of immune checkpoints (p. 18).
7. When conducting pathway enrichment analysis of Lam3+ dendritic cells, they found these cells to be enriched in several pathways (pathways of cytokine-mediated signaling transduction, DC cell differentiation, leukocyte activation, membrane trafficking, and antigen processing and presentation).

However, they conclude that these cells are tolerogenic without giving the substantial proof needed for this conclusion (p.14 + fig. 5i and Extended 5f).

8. It is recommended to validate the interaction between myeloid, regulatory T cells, and NK cells by the use of other cell-interaction algorithms (such as NicheNet and CellPhoneDB) and immunohistochemistry if possible. Further validation is also required for the finding that macrophages have the potential to upregulate the function of regulatory T cells.

9. The statistics are poorly described and sometimes missing, e.g.:

- Inter-patient variation in biologic signatures: is the difference between S135 & S158 vs. S33 & S134 significant? (Extended Fig. 1c-f)
- Comparing % of each cell type between tumor and adjacent tissue: are these differences significant? (Fig. 1e)
- Comparing clonality between CD4 and CD8 T cells (Fig. 4d)

10. Language is sometimes problematic in this manuscript and not consistent, e.g.:

- genes in small vs. capital letters, e.g. p. 14 Lamp3+ vs. LAMP3+
- p. 6 NFkBpathways -> NFkB pathways
- p. 8 suggesteda -> suggested a
- p.15 whilehigh -> while high
- p. 16, 20: prognosis biomarker -> prognostic biomarker
- p. 18: tothe -> to the
- ...

Minor points:

1. The authors compare the percentage of each cell type between tumor and adjacent tissue, but it should be taken into account that this depends on the site of biopsy.
2. Wrong reference to the figure showing the proliferating ability of NK and T cells: Fig. 4i-k -> Fig. 3
3. The studied cohort of seven patients is very heterogenous, consisting patients with ESCC of very diverse stages IIA-IIIB which is another variable that may influence the results.
4. Their written intentions do not always correspond to their actual data. Regarding the CD4 trajectory inference, for example, they state that they will not include regulatory T cells in their analysis, however, this is not consistent with the data shown (Fig 2h: they included cluster 13 which expresses a high level of regulatory T cell genes).
5. The paragraph starting on page 14 includes insufficient description of Fig 5i, but describes the results from extended figure 5f.

Reviewer #3:

Remarks to the Author:

The manuscript from Zheng et al., described the immune landscape in esophageal squamous cell carcinoma (ESCC). The authors profiled stages 2A-3B tumour and peritumoral tissue (5 cm away from tumour) from six male and one female patients. The patients were aged between 54-81 and had not received prior treatment. 3000-9000 cells per sample were analysed from CD45+ cells that were isolated by FACS.

The main limitation of the study is its descriptive nature and with little validation beyond FACS analysis for cytokine production of T cells. Some of the data descriptions were not accompanied by

tissue context or mechanistic understanding of the immune landscape.

1. The study will be significantly enhanced by in situ validation of the differential proportion of immune cells e.g. exhausted T cells, NK cells and macrophage profile in tumour vs peritumour.
2. The cell-cell interaction predictions will be stronger with validation to support the claims Mac-Treg interactions and others. This can be showing the interacting cells in situ and looking at pathways that are up or downregulated in the interacting pairs to support the R:L interactions.
3. The numbering of cell states/clusters is very difficult to follow as they are not necessarily in chronological order – a simplified numbering of CD4_1 to CD4_n in chronological order for the broad immune categories may be easier.
4. The authors can leverage existing scRNA-seq datasets from cancer to annotate and compare myeloid and T cells with other cancers. These can illustrate the similarities and distinctions across cancers which will be more powerful. Zilionis et al., Immunity 2019; Maier et al., Nature 2020 and Zhang et al., Cell 2019 provide good datasets to compare and contrast. Maier and Zhang also describe the regulatory DC subset which the authors observe in their study. Providing this context will add depth to the data presented in this study
5. M1/M2 are in vitro culture states, comparison with macrophages from other cancer data is much more appropriate.
6. Trajectory inference can be misleading as immune cells could have easily been recruited from blood or altered state of resident cells and may not necessarily follow the linear order. For example, tumour associated macrophages may not arise from monocytes alone but be contributed by tissue resident macrophages and this cannot be ascertained in the type of pseudotime analysis. The claims from trajectory inferences need to be tempered or functionally validated as in the linear order.
7. Peritumoral tissue (5 cm from tumour) should not be described as normal but as adjacent or peritumoral tissue due to cancer field effect.
8. What doublet removal step was used in this study – given that the authors used a droplet platform this will be important to assess.
9. Data availability code was not provided
10. Details of what the scale relating to gene expression, scores are not stated on figure panels or not provided in legend for many panels e.g. 1c, 1d, 2b, 2c, 2f (what is expression level?), 5e, 5f, 6a, 6c, 6h, 6i
11. What is 2e gated on? Is this excluding CD4+ T cells as the staining for CD8 looks very discrete.
12. The authors should show the data in supplementary figure of adding anti-PD1 and anti-Lag3 to cultured primary cells on CD8 cytokine production or remove the statement altogether from the MS if stating data not shown.
13. What is the relevance of proliferation data in 3i,j,k
14. The T cell analysis shows very little sharing between tumour and adjacent tissue. This doesn't seem to be clearly stated

Immune suppressive landscape in a human esophageal squamous cell carcinoma microenvironment

Point to point response letter

Reviewer #1:

In this manuscript by Zheng et al, the authors applied 10x single cell technology to obtained RNAseq profile from 7 patients with esophageal SCCs, and additional TCR-sequencing for the T-cell component from these samples. They focused their attention on flow-sorted CD45+ cells only (i.e. immune compartment), from tumor and adjacent normal (more than 5 cm apart) tissue, and therefore managed to extract RNAseq data from a total of 80K+ single cells, surprisingly in some circumstances even getting 9000 cells per sample. They subsequently delve into the different subsets, to derive insights into the immune microenvironment that exists in these cancers which naturally have a poor outcome with little in the way of targeted therapeutic options. In doing, they put together immune micro-environment of ESCC which can be subsequently used to compare either with other tumors of the digestive tract OR with similar histologies (ie SCCs) from other subsites. From a general perspective, this is a well-executed descriptive atlas with an opportunity to shed light on specific details of the immune milieu in ESCC.

However, in studies like these, there is a need to up front decide whether the data is purely descriptive or with functional intent. Supporting experiments can therefore be focused to support the former or latter intent. The main issue in this study that cuts across the entire paper is that it is meant to be descriptive, but there is little to no attempt to validate these findings with supportive data, and that is a critical omission. In contrast, the authors present a single functional experiment (which was not the objective of this paper to begin with), which confounds, rather than illuminates (see below).

Secondly, there is also a need to link some these data in a logical manner in relation to known aspects of tumor immunology, and this is also missing in many sections (see below), and therefore needs to be addressed.

Finally, data about cell-cell interactions (last figure) are speculative at best, and investing such a large proportion of the paper to this, without any functional or supportive data invites criticisms.

We thank the reviewer for carefully reviewing our manuscript and finding our study to be “a well-executed descriptive atlas with an opportunity to shed light on specific details of the immune milieu in ESCC” and we appreciate the reviewer

for pointing out major weakness of our original manuscript. As the first single cell transcriptome analysis on immune cells isolated from ESCC, the main purpose of this study is to provide a descriptive atlas of the tumor infiltrating immune cells in ESCC. Taking the reviewer's suggestion, we conducted a large amount of validation experiments in this revised version of the manuscript. Three different kinds of validation experiments were conducted with the purpose of verifying our findings about major categories of tumor infiltrating cells. First, we obtained additional samples from surgical removed tumor tissues and adjacent tissues and conducted flow cytometry analysis. Second, we conducted staining to provide an in situ look at tumor infiltrating immune cells. These two experiments provide validation for the change of T cells, B cells, DC cells, and monocyte/macrophages between paired tumors and adjacent tissues that was discovered in our original manuscript. Third, we analyzed the bulk tumor RNA-seq data, and used it to infer the major immune cells populations. Please see response to Comment 1.

As the reviewer suggested, we reorganized the entire manuscript and added comparison to known aspects of ESCC immunology in parallel, such as the comparison of Treg cells, exhaustion T cells, CD8 bystanders, macrophage and tolerogenic DC with other reports, please see response below.

Additionally, we verified some interesting findings from cell-cell interaction analysis, e.g. the macrophage and Treg cells interactions through HLA-LILRB1 and IL1 β -IL1R2 by flow cytometry, multi-color IHC staining, co-culture, and antibody blocking experiments, suggesting these L-R interactions may play important role for Treg and macrophage cells function. Importantly, we found that a gene signature derived from WGCNA analysis of monocyte/macrophage was associated with ESCC survival. Taking these findings together, the major conclusions of our single cell transcriptome analyses were supported by our validation experiments and/or analyses. Please see response to Comment 12.

By including all of these results, we provided solid evidence to back our original conclusion. We appreciate the reviewer's comment, which helped us to strengthen our manuscript significantly. Please find more detail about these new results in following responses.

Nonetheless, I feel that if many of these issues can be resolved, this data is still an important reference that warrants it publication. Below I have listed down a number of issues that I feel are important revisions required to be addressed.

We appreciate the reviewer's encouraging comments. We found the comments and suggestions were based on careful and deep consideration. They are clear, specific, constructive, and thus, addressable. Below please find our point-by-point

response. The studies and modifications suggested by the reviewer significantly enhanced our manuscript. We thank the reviewer for the help.

1. One of the major claims in Figure 1 and 2 are the proportions of the different immune cell lineages, comparing tumor versus normal and across different patients. While the data for these differences are interesting, it needs to be backed up by other methodologies or orthogonal data- flow cytometry, IHC, bulk sequencing of unprocessed samples. The main issue is that processing time and methodology can greatly alter specific lineages, and these numbers may be skewed between and across experiments. Neutrophils and macrophages are especially vulnerable in these situations, and statements about proportions of these need to be validated before being stated as fact. Even across all the normal samples, there is significant variation using scRNAseq. The data shown here could even be validated against published bulk RNAseq data comparing tumor vs normal.

We thank the reviewer for the suggestion and agree with that validation is required to make solid conclusions in our study. During revision, we have conducted a large amount of validation experiments and analyses.

Firstly, we collected additional samples with paired tumor and adjacent tissues, followed by staining with antibodies against markers of major immune cell types, such as CD3 (T cell), CD56 (NK), CD20 (B cell), CD68 (macrophage), MPO (neutrophil). Consistent with the single cell RNA-seq data, we found an increase of T cells and macrophage and a decrease of NK and B cells in tumor tissues, compared to adjacent tissues. Representative images and quantifications are shown in a newly generated Supplementary Fig. 4.

Second, we conducted flow cytometry analyses of samples for eight major cell types, including the five types analyzed by staining. Consistent with the staining results, we found T cells and macrophage were increased in tumor tissues while NK and B cells were decreased. Additionally, tumors also carried more DC cells than adjacent tissues. These data are shown in a new Supplementary Figure 5.

Thirdly, we tried to extract immune cell infiltration information from bulk RNA-seq data. We applied CIBERSORT to the TCGA ESCC dataset. Some results are consistent with our findings, like decreasing B cells, elevating monocyte/macrophages and DC in tumor tissue. However, the percentage of T cells and NK cells are inconsistent with our findings (Response Fig.1). One possibility is that there were 77 tumor tissues and only 11 normal tissues in the TCGA dataset. Because of the heterogeneity among patients, most unmatched samples in the dataset may produce unreliable

results. Additionally, inferring immune cell infiltration by bulk RNA-seq data still present a challenge and is largely dependent on the algorithms.

In our primary scRNA-seq analysis, we did not identify a significant population of neutrophils. As the reviewer pointed out, the processing time and methodology of single cell transcriptome analysis can greatly alter some vulnerable lineages, like neutrophil and macrophage. In our IHC staining and FACS analyses of new pairs of tumors, we did detect a small percentage of neutrophil. IHC staining of MPO showed 0-44 positively stained cells per field in adjacent tissue, and 3-46 positive cells in tumor. Flow cytometry also indicated low proportion of neutrophils ($1.62 \pm 0.47\%$ in adjacent tissue and $1.74 \pm 0.63\%$ in tumor tissue in all CD45+ cells). Thus, neutrophils were relatively rare but did exist in the ESCC tissues. The failure of detecting neutrophils in our scRNA-seq was most likely caused by limitations of the 10x Genomics technology used in our study. Neutrophils are usually significantly under-represented regarding their identification and characterization, especially for tumors isolated from human patients¹⁻⁴. This could be caused by (1) the relatively long process of surgical resection of human tumors, single cell isolation, and sequencing; and 2) neutrophils have very low RNA content and contain high activity of RNases, which could potentially result in fewer transcripts being detected and less usable sequencing reads; 3) during scRNA-seq data analysis, low-quality cells (<400 genes/cell and >10% mitochondrial genes) were excluded, so these may affect the identification of neutrophils. On the other hand, macrophage was easier to detect in our scRNA-seq processing. We have included this discussion in the revised manuscript (Page 7).

By both staining and performing flow cytometry on the additional samples, or analysis the bulk RNA-seq data, we observed a significant degree of variation of infiltrating immune cells, even across adjacent tissues, which was consistent with our scRNA-seq data.

Supplementary Fig. 4: IHC stained of CD3 (T cell), CD56 (NK), CD20 (B cell), CD68 (macrophage), MPO (neutrophil) in adjacent and tumor tissue of ESCC (n=12), and analysis of positive of cell number in each field. * $P < 0.05$; two-sided t-test.

Supplementary Fig. 5: FACS analysis of adjacent and ESCC tumor tissues (n=12) of immune subsets, including T cells, B cells, plasma, NK, DC, mast, neutrophils, and macrophages, the gating strategies (a) and percentages (b) were shown.

* $P < 0.05$, ** $P < 0.01$; two-sided t-test.

Response Fig. 1. Analysis of bulk RNA-seq data of ESCC from TCGA (Adjacent tissue n=11, tumor tissue n=74), the percentage of immune cell subsets analysis from CIBERSORT. ** $P < 0.01$; two-sided t-test.

2. *There is insufficient data about tumors-normal pairs, especially given the variation seen across adjacent normal...were there anything specific about those with higher T cells compared to the rest? Were they closer to the tumor? I wouldn't like to see a scaled diagram or cartoon showing the location of each tumor and normal pair.*

As the reviewer pointed that size of the sample set used for single-cell discovery of T cell heterogeneity was limited. During tissues collection, we tried to avoid potential cross contamination between tumor and adjacent tissues, by harvesting matched adjacent samples from the opposite end of the removed tissues. All adjacent tissues were at least 5 cm away from the closest tumor tissues. The difference on distance to tumor tissues was likely not responsible for the variation across adjacent tissues. We now included a cartoon (Revised Fig.1a) and one sample photo taken during tissue collection to indicate location of tumor and adjacent tissues (Response Fig. 2). As the reviewer 3 suggested, peri-tumoral tissue (5 cm from tumor) should not be described as normal but as adjacent or peri-tumoral tissue due to cancer field effect. We have changed normal tissue to tumor adjacent tissue in the revised manuscript.

As the reviewer noticed, we observed variations among adjacent tissues, e.g. the proportions of T cells were 29.04%-64.65%, N=7. Staining and flow cytometry analyses of additional 12 tumor-adjacent pairs gave similar levels of variations (Supplementary Fig. 4 & 5). However, PCA analysis of immune profiles clustered the adjacent tissue together (Revised Fig. 1h). Additionally, we analyzed the TCGA bulk RNA-seq data of ESCC, and found that the eleven normal tissues

showed similar variation, compared to our scRNA-seq data. For example, T estimated cell proportion was 10.84% - 50.39%, N=11 (Response's Figure 1). The large variation of tumor infiltrating immune cells in ESCC adjacent tissues may be caused by patient heterogeneity, local inflammation, and/or potential esophageal infection.

Revised Fig. 1a: Schematic diagram of the experimental design and analysis.

Response Fig.2. The position of tumor (red circle) and adjacent tissue (yellow circle) in one representative sample.

3. There is little to no data about the associated tumor in each case as well? Do you know the tumor fractions of the region used for these above analyses? Were they at the middle or periphery of the cancers? There is no profiling data of the tumor cells presented.

Tumor biopsies were collected from the resection part of the tumor tissue that included middle and periphery of the cancers. As shown in Reviewer Fig. 2, the boundaries of tumor tissues were clear, and thus, the contamination of adjacent tissue was unlikely. Besides processing for scRNA-seq and TCR-seq analysis, a part of the biopsies was fixed and histologically analyzed. Based on H&E staining images, cancer cells dominated the tumor region. We included representative images of all seven tissues in a new Supplementary Fig.1.

Additionally, we further analyzed non-immune cells clustered as “other” from clustering analysis of single cell RNA-seq data (Supplementary Fig.2). We found that most of them from tumor tissue had copy number variations (CNV), including

both amplifications and deletions, suggesting this cluster was dominated by tumor cells. Some CNVs are known to be amongst the most frequently gained in tumors, e.g. gain of chr1q21, 3q and 8q based on a pan-cancer CNV study⁵. This study primarily focuses on characteristic of tumor infiltrating immune cells in ESCC. A follow-up study is currently undergoing in our laboratory to examine the interaction between tumor cells and immune cells in ESCC, including how neoantigens affect T cell-mediated cytotoxicity and tolerance.

Supplementary Fig.1. HE staining showing tumor cells of seven patients with ESCC, showed the 1.0x, 10x and 40X, respectively

Supplementary Fig. 2. Chromosomal landscape of inferred large-scale copy number variations (CNVs) distinguishes malignant (lower panel) from immune cells (up panel). The “other” cell cluster was shown with individual cells (y-axis) and chromosomal regions (x-axis). Amplifications (red) or deletions (blue) were inferred by averaging expression over 100-gene stretches on the respective chromosomes.

4. I do not understand the part of data that states “We found inter-patient variation in biologic signatures, including hypoxia, epithelial-mesenchymal transition, inflammation response, and TNFA-via NFKB pathways”, when these were based on profiling immune cells...would you expect EMT or hypoxia in these cells?

We apologize for not interpreting the results clearly and accurately. We performed gene set enrichment analyses and attempt to interpret biological signals that were in the studied immune cells. The cluster median of each gene was taken per patient, and the cluster medians were z-scored across patients. The z-scored values were plotted as a heatmap, facilitating a comparison of signatures across patients. The results of a number of these signatures such as hypoxia, epithelial mesenchymal transition, inflammatory response, and TNFA-signaling via NFKB were showed in the original manuscript, in order to show various cell states across patients. As the reviewer correctly pointed out, some of these data were lacking for an explanatory biological basis. For example, immune cells were not likely undergoing EMT. Some genes in the EMT signature were also important for immune responses, such as TGFβ1, TGFβR3, CD44, and IL6. Thus, the results may reflect variations that are not EMT, but other biological processes sharing some of the genes with EMT. This figure was intended to show phenotypical heterogeneity amongst tumors, but instead, it caused confusion. In the revised version, we removed the EMT signature from the analysis. On the other hand, tumor infiltrating immune cells may

undergo hypoxia, since they are located in a microenvironment that was usually hypoxic⁶.

5. Data on the T cell compartment was described in detail but I felt that the links and connections made within this dataset, and what is known in the literature very scanty. For example, demonstrating that cytotoxic cells were more likely present at the adjacent normal regions is an important point, but was not elaborated further in the context of other cancers which describe these as being within the tumor as bystanders. Similarly the clusters of exhausted cells were well described, but there was no proper connection made between these. The use of monocle in this context is also somewhat inappropriate. Monocle is used to demonstrate evolution of a cell type or cell state transitions and there is no evidence shown that the CD8 cells are derived from a common origin. In fact the authors are unable to demonstrate a convincing naïve population in their tumors at all!

We appreciate the reviewer's suggestion of enhancing the connection between our data on T cell compartment with more literature.

The ratio of cytotoxic T cells in adjacent tissues was much higher than in tumors (Revised Fig. 3d and 3e), despite the decrease of total infiltrating T cells in adjacent tissues (Revised Fig. 1i). It was consistent with the recently published paper⁷. The target specificity of these cytotoxic T cells, e.g. being tumor specific or non-tumor specific, was not further analyzed in our original manuscript. Dr. Simoni and his colleagues reported that bystander CD8+T cells that recognized cancer unrelated epitopes lacked expression of CD39⁸. Thus, we analyzed the CD39 expression in CD8 T cells in different clusters isolated from tumors and adjacent tissues. Consistent with the original report, CD39 expression was higher in pre-exhausted and exhausted CD8 T cells (CD8-C5-CCL5, CD8-C6-STMN1, and CD8-C7-TIGIT). Additionally, there was an increased proportion of CD39⁻ cell in adjacent tissues, indicating a large fragment of these infiltrated CD8 T cells were bystanders that were possibly specific to other antigens, e.g. Epstein-Barr virus, human cytomegalovirus or influenza virus. These data were included in revised Fig. 2j.

We identified three clusters (CD8-C5-CCL5, CD8-C6-STMN1, and CD8-C7-TIGIT) of CD8 T cells expressing different levels of exhaustion markers, including PD-1, TIGIT, CTLA-4, TIM-3, and LAG-3. Distinguished stages of CD8 T cell exhaustion have been reported by studies in other cancer types, for examples, early (PD-1-int, Eomes-low) vs and terminal (PD-1-hi, Eomes-hi) exhaustion⁹. We checked the expression of PD1 (PDCD1) and Eomes expression in these clusters. We found that CD8-C5-CCL5 expressed a lower level of PDCD1 and EOMES compared with CD8-C7-TIGIT, while

CD8-C6-STMN1 was in the middle. It suggested that CD8-C5-CCL5 cells were at an early stage of exhaustion, while CD8-C7-TIGIT was terminal exhausted. CD8-C6-STMN1 was likely in a transition stage between CD8-C5-CCL5 and CD8-C7-TIGIT. The exhaustion at the reversible stage can be rescued by checkpoint blockade, but not at the permanent stage that is established by epigenetic alterations^{10,11}. We checked the expression levels of TOX and NFATC2; both genes participate in establishing epigenetic programs to install permanent exhaustion status in CD8 T cells. The revised Fig. 2i shown that CD8-C7-TIGIT expressed a higher level of TOX and NFATC2, suggesting the terminal exhaustion state.

As the reviewer noticed, we did not find a significant population of naïve CD8 T cells in our samples. Consistently, the naïve gene signature was largely absent in all CD8 T cell clusters (Revised Fig. 2b and 2d). Similar studies in other cancer types, e.g. melanoma and liver carcinoma, identified a limited number of naïve CD8 T cells^{12,13}. In contrast, we identified a clearly naïve CD4 T cells cluster (CD4-C1-CCR7) (Revised Fig. 2b, and 2c). It is possible that naïve CD8 T cells less frequently infiltrate into the esophagus or are activated by the local environment, making them less likely to be detect as an independent population. Additional studies are needed to validate this phenomenon and identify the mechanism. We included this in the discussion section of the updated manuscript (Page21-22).

We agree with the reviewer that Monocle is usually used to demonstrate evolution of cell types or transitions of cell states. CD8 T cell trajectory analysis showed that each cluster aggregated based on expression similarities. CD8-C1-NKG7, the cytotoxic cluster was aggregated on the opposite side of the exhausted cluster, CD8-C7-TIGIT. Other clusters were mostly located between them, presented a continuous trajectory from cytotoxic to exhaustion state, suggesting the transcriptional gradients contribute to T cell heterogeneity. In order to verify our results, we applied another algorithm, Slingshot¹⁴ to our data and obtained comparable results (Revised supplementary Fig.6g). We agree with the reviewer that tumor infiltrating CD8 cells are not derived from a common origin in tumors. However, the monocle analysis here still provides useful information about transitions of cell states, which exist in tumors.

Revised Fig.2j: Violin plot showed the ENTPD1 expression in CD8 T cell clusters and 7 ESCC and adjacent samples. The expression was measured as the $\log_2(\text{count}+1)$.

Revised Fig. 2i: Violin plot showed the PDCD1, NFATC2, EOMES and TOX in CD8-C5-CCL5, CD8-C6-STMN1, and CD8-C7-TIGIT clusters. The expression was measured as the $\log_2(\text{count}+1)$.

Supplementary Fig.6g: The trajectory of CD8 T cells state transition in a two-dimensional state-space inferred by Slingshot. Each dot corresponds to one single-cell, colored according to its cluster label. Arrows showed the increasing directions of certain T cell properties.

6. Monocle data for the CD4 populations is even more illogical, as these are not subpopulations that transition from one to another in the tumor. Are you suggesting that Tregs and Th cells are matured in the tumor? If not, it does not make sense to align this in the same trajectory. There is little biological basis to use what is a powerful tool.

We apologize for not clearly explaining the trajectories of CD4 T cells in the previous manuscript. Treg cells (CD4-C6-FOXP3) was excluded from the monocle analysis of CD4 cell (Revised Fig. 2e). CD4-C5-STMN1 express low FOXP3, however, it was clustered as exhausted CD4 T cells, based on marker gene and a higher exhaustion signature. Thus, the trajectory was built for Th cells only. The inferred development trajectory exhibited a branched structure. CD4-C1-CCR7 was characterized by the naïve state and positioned at the opposite end of exhaustion clusters (CD4-C5-STMN1). CD4-C4-IFIT3 represented the cytotoxic CD4 T cells and was located at another branch; CD4-C2-TCF7 and CD4-C3-CD40LG were located in between, indicating their intermediate states. We used Slingshot method to construct the trajectory and observed a similar trend with naïve CD4-C1-CCR7 and exhausted CD4-C5-STMN1 localized on two opposite ends of the trajectories. The trajectory generated by Slingshot has been included in revised Supplementary Fig.6f.

Supplementary Fig.6f: The trajectory of Th cells state transition in a two-dimensional state-space inferred by Slingshot. Each dot corresponds to one single-cell, colored according to its cluster label. Arrows showed the increasing directions of certain T cell properties.

7. *I found that the functional experiment introduced in figure 3e a bit random, not to mention the fact that the most important conclusion of increased TNF-alpha after CTLA4 blockade, was unconvincing and barely reached statistical significance. Moreover, the lack of activation of IFNG and GZMB is also not supportive of their conclusion that this blockade targets Tregs, which then affects CD8 cells. There are so many missing steps to draw this conclusion presented here, and this can either be addressed or omitted completely.*

We agree with the reviewer that this experiment is not closely related to this study and the data are far from drawing a solid conclusion. We have removed this result in the revised manuscript. We thank for the reviewer's suggestion.

8. *There are lineage connections within the CD4 and CD8 populations that can be drawn but these have not been highlighted or presented here at all.*

We agree with the reviewer's comment and thank you for this suggestion. In the revised manuscript, we further extend of our analysis to the lineage connection within the CD4 and CD8 populations, respectively. We found CD8+T cells form a gradient of transcriptional states within tumors, including CD8-C1-NKG7, the cytotoxic effector pool, CD8-C3-GZMK, a transitional CD8 effector T pool, and C5, C6, and C7, three clusters of pre-exhaustion and exhaustion CD8 T cells, marked by expression of immune checkpoint molecules such as TIM3, PD-1 and LAG3. However, exhaustion T cell pool does not form a discrete cell population but is part of a wide differentiation spectrum, spanning from early exhaustion toward terminally exhaustion T cells. More interestingly, a strong clonal sharing of pre-exhaustion and exhaustion CD8

cells was observed, but not with cytotoxic CD8 cells. In CD4 populations, some of the gene activated in Tregs (CD4-C6-FOXP3) overlapped with those characteristic of the exhaustion program in CD4 Th cells (CD4-C5-STMN1). We compared gene expression enrichment in both T cell types and compared them to the naïve like cell population (CD4-C1-CCR7). Genes enriched in both the exhaustion and the Treg cells included regulatory molecules and many co-inhibitory and co-stimulatory receptors (e.g., TNFRSF9, CSF1, IL1R2, and TIGIT). In contrast, CD4-C6-FOXP3 expressed much higher levels of FOXP3, IL2RA, and CTLA4 than CD4-C5-STMN1, while CD4-C5-STMN1 expressed high levels of TNF, TOX, PDCD1, IFN, CXCL13 and CD38. Visualization of the exhaustion and Treg scores confirmed the overlap between these two differentiation programs (Revised Fig. 2c and 2g, Supplementary Fig. 6h). And interestingly, exhaustion T cells including CD4 and CD8 were the major intratumoral proliferating immune cell population (Revised Fig. 3k and 3l).

As the reviewer recommended, clonal amplification of cells carrying identical TCR sequences across clusters was a strong evidence to support the connection among these clusters and indicated the transition of cell status. While most cells contained unique TCRs, clonal amplification was observed at varying degrees in different clusters. Indeed, we found sharing of TCR sequences among all clusters in CD4 cells, including Treg, and all clusters within CD8 cells, with the exception of C2 (Revised Fig.4g and h). These data suggested a broad differentiation after T cell priming. It was notable that exhausted CD8 T cells (CD8-C7-TIGIT) carried a higher percentage of shared clones with other CD8 clusters, especially with the pre-exhaustion clusters CD8-C5-CCL5 and CD8-C6-STMN1 (Revised Fig.4i and Supplementary Fig.8b). It was consistent with the related status of these clusters and the multi-step exhaustion hypothesis. In CD4 T cells, CD4-C6-FOXP3 Treg cluster displayed 14.4% of shared clonotypes with naïve cluster CD4-C1-CCR7 in tumors and 40.7% with the adjacent tissue, suggesting a high frequency of Treg cells were derived from naïve CD4 T cells and locally expanded (Revised Fig. 4j and Supplementary Fig.8c).

We appreciate for the reviewer's constructive suggestion. Please see the response to comment 8. Indeed, the clonal identifiers obtained by TCR analysis provide us a unique dataset to infer the lineage structure of T cells in tumors.

10. What proportion of T cell did they have TCR data for? It is difficult to make comments about TCR clones and clustering, and in fact this may be a biased analyses of better quality cells rather than distribution of TCR-clonality. It is intriguing that cytotoxic bystander cells in the adjacent normal tissue were more clonal compared to cells within the tumor itself, can the authors speculate why this might be so? Or if this is a phenomenon relevant to a contaminated site like the esophagus. Unfortunately the lack of attempt to link this data makes this section extremely descriptive with little or no major conclusions drawn.

We apologize for not providing enough information about the TCR data. In brief, we recovered TCR α and TCR β sequences from 26,920 and 31,440 T cells, respectively. The ratios of recovering full length α and β chains were 70.59% (26,920/38,134) and 82.45% (31,440/38,134), respectively. The proportion of T cells having both full length chains was 69.94% (26,672/38,134), similar to previous reports¹⁵.

We agree with the reviewer that the enrichment of better quality cells during the sequencing process may introduce bias on TCR clonotype analysis. The 10X Genomics single-cell VDJ sequencing is a relatively new technique, but has been widely used to analyze TCR clones and clusters^{2,12}. The bias towards high quality cells cannot be completely avoided in the current experimental setting. However, it is unlikely to affect the major conclusion of our study for the following two reasons: (1) the majorities of cells were high quality: the ratios of recovering full length α and β chains were about 70%. (2) It is unlikely that cells carrying different TCR clonotypes behaved differently during the sequencing process, leading to different success rates of TCR retrieval. However, we cannot fully exclude the possibility that the bias to high quality cells affected the TCR clonotype analysis. Technique advances are required to improve TCR clonotype analysis at single cell level. The variable TCR clonal type composition and their differential expression of key gene signatures suggested the functional states of and relationship among these clusters.

We found the clonality of T cells as a whole in the paired tumor and adjacent tissues were varied (Revised Fig. 4b). In some samples, such as S150, there were more clonal amplified T cells in tumor, compared to adjacent tissue. Other samples, such as S135, T cells were less clonal in tumors. However, if we focus on certain clusters of T cells, there were clear trends. For example, Treg (CD4-C6-FOXP3) showed dramatic increase of clonality in tumors, compared to match

adjacent tissues in most patients (Revised Fig. 3f). It suggests active proliferation of Treg in tumors. In contrast, CD8-C1-NKG7, the most cytotoxic CD8 T cell cluster, had significantly less clonal T cells in tumor, compared to adjacent tissues (Revised Fig. 3e), as the reviewer pointed out. It suggests the inhibition of cellular proliferation of cytotoxic CD8 T cells in the tumor microenvironment. This may contribute to the decreased numbers of tumor infiltrating effective CD8 T cells and to the immune suppressive microenvironment. A similar finding was recently reported in bladder cancer, in which cytotoxic CD8 T cells (FGFBP2+ cluster in their analysis) were more clonal in normal tissues than tumor tissues⁷. On the other hand, it is possible that the clonal amplification of esophagus cytotoxic cells was due to the occasional exposure to non-tumor antigens in esophagus, that most CD8-C1-NKG7 cells in adjacent tissues express low level of CD39, indicating they are bystander CD8 T cells. As a part of the digestive tract, the esophagus has an increased chance of getting contamination with bacteria and viruses. Consistently, there were very few sharing of clonal TCR sequences in CD8-C1-NKG7 between tumor and normal tissues (Supplementary Fig.8d).

11. Analyses of the myeloid component was probably the highlight of this paper and the data most intriguing. Again, I would have liked to have seen supportive flow cytometry data, especially to address the missing neutrophil populations herein. Trajectory analyses was also important and appropriate (for once) showing directionality from immature monocytes to M1 or M2, however I was unsure if Figure 5c was showing all the myeloid cells or only cluster 1,5,6,8? And if so, why and how were these chosen? Again Scenic analyses was appropriate here, but the authors did not link these with downstream genes through network analyses nor associate this data with previous publications. Data on DCs were also descriptive.

We thank the reviewer for highlighting our analysis on myeloid linkages.

As the reviewer suggested, we did flow cytometry analysis and IHC staining on additional samples and determined the proportions of major immune cell types, including myeloid lineages in ESCC tumors. Please see the response to Comment 1 for more information. Specifically, we did detect a small fraction of neutrophil in tumor and adjacent tissues. IHC staining of MPO showed 0-44 positive stained cells per field in adjacent tissue, and 3-46 positive cells in tumor. Flow cytometry indicated a low proportion of neutrophils ($1.62 \pm 0.47\%$ in adjacent tissue and $1.74 \pm 0.63\%$ in tumor tissue in all CD45+ cells). Thus, neutrophils existed but were relatively rare in ESCC tissues. The low number of neutrophils plus the technique difficulties (discussed in detail in the response to Comment 1) for profiling neutrophils in single cell RNA-

seq likely led to the number of captured neutrophils was below the threshold for being clustered independently. The validation and discussion is included in the revised manuscript (Page 7).

We apologize for not clearly describing the data in the trajectory analysis of myeloid cells. In Figure 5, the analysis was conducted with Mono-C1-VCAN, Macro-C3-CSF1, Macro-C1-IL6, Macro-C2-IL1RN four clusters. By using published gene signatures of the monocyte, M1, M2 and MDSC, we found that Mono-C1-VCAN showed a strong monocyte signature, Macro-C1-IL6 and C1-CSF1-Maro represented M1 and M2 macrophage stage, respectively, and C6-ILRN-Macro had enriched monocyte, M1 and M2 signatures. Thus, these clusters may represent classic difference of monocyte/macrophage stages in tissue. Mono-C2-IL1B, Mono-C3-FOS showed an intermediate state; MDSC-C1-C1QC and MDSC-C2-APOE clusters were likely to be MDSC, and Macro-C4-LILRB2, preferential and enrichment in adjacent mucosa versus tumors were denoted as tissue resident macrophages. As the reviewer 3 suggested, immune cells could have easily been recruited from blood or altered state of resident cells and may not necessarily follow the linear order. Thus, we picked four relatively typical clusters that represents the monocyte, M1, M2 and intermediate state, and did the trajectory analysis, data showed the increasing directions of certain cell properties, may represent the cell state transitions. We applied SCENIC to explore the transcription factors that may regulate monocyte, M1 and M2, development. The transcriptional factor BHLHE40 was specifically expressed in M2. We conducted the network analyses to identify the BHLHE40 downstream genes and analyze the functions through Metascape (Revised Fig.5j). Data showed that BHLHE40 target genes were associated with myeloid cell differentiation and negative regulation of cellular response (Revised Supplementary Fig.9i). BHLHE40 has recently been reported to mediate tissue-specific control of macrophage self-renewal, proliferation (31061528) and we found BHELHE40 was associated with poor prognosis of ESCC (Supplementary Fig.9j). Bheleh40 may play critical role in inducing TAM toward M2 phenotype, further studies is needed to explore the detail mechanisms.

To dive deeper into tumor infiltrating dendritic cells, we further analyzed the tolerogenic DC in additional samples by flow cytometry and multi-color IHC staining. Data showed that LAMP3+DC expressed much higher CD83, CCR7 and PDL1 than LAMP3-DC (Revised Fig. 6e and f), suggesting the maturation, migration and regulation ability of LAMP3+DC. IHC staining also validated the existence of tolerogenic DC that express CD11C, LAMP3, PDL1, and IDO (Revised Fig. 6g). We further treated DC with IFN γ and LPS. Interestingly, we found IFN γ and LPS stimulation induced

PDL1 and IDO expressing in DC (Fig. 6h) and had increased ability to induce FOXP3 expression when co-cultured with CD4+CD45RA+ naïve T cells (Fig. 6i). These data suggested that IFN γ and LPS may induce tolerogenic DC *in vitro*. All these data suggested functional importance of tDC cells and have been included in the revised manuscript (Page17-18).

Revised Fig. 5j: Network analysis of transcription factor BHLHE40 and its target genes through iGraph, the size of the circle represented the co-expression relationship score.

Supplementary Fig. 9i: Bar graph of enriched terms of BHLHE40 target genes generated by SCENIC of Fig.5h.

Revise Fig. 6e-i: (e) FACS analysis of LAMP3 and PDL1 expression in DC cells from adjacent and ESCC tumor tissues (n=5), ** $P < 0.01$, pair two-sided t-test. (f) CCR7 and CD83 expression in LAMP3+DC and LAMP3- DC cells from adjacent and ESCC tumor tissues by FACS (n=4), * $P < 0.05$, pair two-sided t-test. (g) Detection of CD11C+LAMP3+PDL1+IDO+ DC in ESCC tissue by multi-color of IHC. (h) DC cells were stimulated with LPS and IFN γ or not for 24 hours, PDL1, LAMP3 and IDO expression was analyzed. One of the three experiments was represent, error bars representing \pm SEM, * $P < 0.05$, two-sided t-test. (i) DC cells were stimulated with LPS and IFN γ or not for 24h and cultured with CD4+CD45RA+ naïve T cells for 4 days, FOXP3 expression were measured. One of the three experiments was represent, error bars representing \pm SEM, * $P < 0.05$, two-sided t-test.

12. The last section of the paper was probably the most speculative. Many of these associations are known and the flaw of the methodology used only analyzes expression levels of ligand receptor pairs. I would have liked to have seen specific analyses of downstream and pathway analyses of interesting pathways that the authors were trying to highlight, if not this is just a long meandering speculative section.

We appreciate the reviewer's comment. The communication between different subpopulation of immune cells, the majority being through ligand-receptor interactions and these play key roles in determining the tumor microenvironment and response to therapeutics. Thus, we analyzed our single cell RNA-seq data by an approach developed for this purpose¹⁶. This type of analyses have been applied in other single cell sequencing studies^{4,17,18}. As the reviewer pointed out, the flaw of the methodology is basically based on the expression levels of ligand/receptor pairs. Thus, identified interactions need to be validated by independent methods and functional experiments.

First, we applied scTHI (<https://github.com/miccec/scTHI>), another widely used method to predict the interaction between Treg cells and macrophage, and obtained comparable results (Supplementary table 8). We then focused on two interactions that had high interaction scores in both algorithms and potential contributions to the immune suppressive microenvironment: (1) IL-1 β in macrophage and its receptor IL1R2 in Treg cells and (2) MHC in Tregs and LILRB1 in macrophage.

To verify the interaction between macrophage and Treg introduced by IL-1 β /IL1R2, we first detected IL1R2 expression in ESCC and adjacent tissue through flow cytometry in Treg cells. We found Treg cells in tumor tissues had increased IL1R2 expression (Revised Supplementary Fig.11a). Multi-color IHC staining also validated the IL1R2 expression in Treg

cells (Revised Fig.7d). It had been reported that IL1R2, as an IL1 β decoy receptor, binds to IL1 β and blocks IL1 β dependent activation of follicular helper T cells¹⁹, so we then explored the IL1R2 function in Treg cells by co-culture and antibody blocking experiments in vitro. Data showed that IL1 β could stimulate CD4+CD25- Teff cells proliferation and IFN γ production compared to Teff cell alone. The addition of Treg cells markedly inhibited this IL-1 β -dependent Teff cells activation. However, when adding IL1R2 antibody in the co-culture system, Treg cells ability to inhibit effect T cells activation decreased (Fig 7e and f, Supplementary Fig. 11b and c). These data suggested that IL1R2 expressed on Treg cells may enhance Treg cells function and block IL1 β dependent effector T cells activation. It has been reported that IL1R2 expressed on activated tumor Tregs and correlated with poor prognosis in lung adenocarcinoma^{20,21}.

We also predicted interaction between MHC in Treg and LILRB1 in macrophages (Fig. 7g). The MHC receptor LILRB1 is the negative regulators of myeloid cell activation and promotes M2 suppressive state. Engagement of MHC class I by the inhibitory receptor LILRB1 suppresses macrophages and was a target of cancer immunotherapy²². This pathway between Treg and macrophage was not clearly demonstrated. So first, we analyzed LILRB1 expression in macrophage by single cell RNA-seq and further detected expression by FACS. Data showed that LILRB1 expression increased in ESCC macrophages than adjacent tissue (Revised Fig. 7h and I, supplementary Fig.11d). Treg cells expressed high level of HLA-A, HLA-B and HLA-C (Supplementary Fig. 11e and 11f). Multi-color IHC staining also showed the potential physical interaction (co-localization) between LILRB1 expression macrophages and Treg cells (Fig. 7j). Our *in vitro* study found that Treg cells promoted macrophage toward M2 phenotype differentiation, inducing CD163 and PDL1 expression, and decreasing TNF α expression. However, after blocked LILRB1, CD163 and PDL1 expression decreased, and TNF α expression increased compared to IgG control (Fig. 7k and l). These data suggested that Treg cells could modulate macrophage function through HLA-A, B, C and LILRB1 interaction, blocking this pathway may promote antitumor immunity in ESCC.

Revised Fig.7 d, e, and f: (d) Detection of CD4, FOXP3 and IL1R2 expression in ESCC tissue by multi-color IHC staining. (e, f) CD4+CD25⁻ (Teff) cells were stimulated with IL1 β or not, and Treg cells cocultured with Teff cells in the anti-IL1R2 or IgG antibody for 72h, the dividing of Teff cells (e) and IFN γ expression in Teff cells (f) was measured by FACS.

Supplementary Fig. 11a, b and c (a) FACS analysis of IL1R2 expression in Treg cells from adjacent and ESCC tumor tissues (n=5). (b) Bar plot showed Teff dividing cell in Fig.7e, error bars representing \pm SEM, * $P < 0.05$, ** $P < 0.01$, paired Student's t-test. (c) Bar plot showed Teff expressing IFN γ cell in Fig.7f, error bars representing \pm SEM, * $P < 0.05$, paired Student's t-test.

Revised Fig.7 h, i, j, k, and l: (h) Violin plot showed the LILRB1 average expression in macrophage from tumor and adjacent tissue. The expression was measured as the $\log_2(\text{count}+1)$. (i) FACS analysis of LILRB1 expression in CD68+ macrophage cells from adjacent and ESCC tumor tissues ($n=5$). (j) Detection of CD4, FOXP3, CD68 and LILRB1 in ESCC tissue by multi-color IHC. (k, l) Macrophage were co-cultured with Treg cells or not, flow cytometry measured CD163, CD206, PDL1 (k) and TNF α (l) expression in CD68 macrophage cells in the presence of anti-LILRB1 or control IgG antibody.

Supplementary Fig.11d, e, f, g and h (d) Bar plot showed macrophage cells LILRB1 MFI in tumor and adjacent tissue in Fig.7i, (n=5), * $P < 0.05$, paired Student's t-test.(e)Violin plot showed the HLA-A, HLA-B, HLA-C average expression in Treg cells from tumor and adjacent tissue. The expression was measured as the log2 (TPM+1). (f) FACS analysis of HLA-A, B, C expression in Treg cell from adjacent and ESCC tumor tissues (n=5). (g) Bar plot showed macrophage cells expressing CD163,CD206 and PDL1 in Fig.7k,error bars representing \pm SEM, * $P < 0.05$, paired Student's t-test. (h) Bar plot showed macrophage cells expressing TNF α in Fig.7l, error bars representing \pm SEM, * $P < 0.05$, paired Student's t-test.

13. Even the survival analyses based on cell-cell interaction was rather mundane. They just picked a macrophage M2 signature and showed its expression levels were prognostic in ESCC (lowish p value) and the in a bunch of other cancers which have no similarity to ESCC. The least I expected here was to use a composite signature that took into account co-expression models to extract this data and apply this across other SCCs where the conclusion actually makes biological sense!

We thank the reviewer for the suggestion. In order to select a composite signature, we use WGCNA to analyze gene expression data of single cells from monocyte/macrophages for gene correlation. This demonstrated different levels of similarity between monocyte/macrophage clusters. Interestingly, we found the Turquoise module was positively

correlated with Mono-C2-IL1B and Mono-C1-VCAN, the monocyte clusters, and negatively correlated with M2 cluster Macro-C3-CSF1, and MDSC clusters MDSC-C1-C1QC, MDSC-C2-APOE (Revised Fig.5d and e). These module genes were associated with myeloid leukocyte activation, and activation of immune response (Supplementary Fig. 9f). We further analyzed the genes in this module and their association with Mono-C1-VCAN (Fig. 5f), and selected the most correlated top 50 genes to form a signature (Supplementary table 8) and applied this signature to TCGA-ESCC data. The patients with a high expression of this signature had significantly better prognosis, compared to the patients with a low expression of the signature (Fig. 5g). A similar association was observed in other tumor types with similar pathology, such as cervical squamous cell carcinoma and lung squamous cell carcinoma (Supplementary Fig. 9g). This suggested that the signature may serve as a prognosis biomarker in ESCC and other squamous cell carcinomas. We thank the reviewer's constructive suggestion.

Revised Fig. 5d, e, f, g: (d) A gene co-expression network of single cell from monocyte/macrophages was constructed by weighted correlation network analysis. The heatmap shows the topological overlap matrix among all genes used in the analysis. The darker color represents a higher overlap. The hierarchical clustering and module assignment of genes were shown along the left side and top. (e) Modules-trait relationships in all single cell sub-clusters. Within each heatmap, red indicates a positive correlation and green indicates a negative correlation. The numbers in brackets were correlation P values. (f) The turquoise module membership and gene significance for Mono-C1-VCAN. The most correlated genes were labeled. (g) The Kaplan-Meier overall survival curves of TCGA ESCC patients with top most correlated 50 genes generated in revised Fig.5f.

Supplementary Fig. 9f and g: (f) Bar graph of enriched terms generated in the Metascape website across the input gene lists generated in Fig. 5d Turquoise module. (g) The Kaplan-Meier overall survival curves of TCGA CEVS and LUSC patients with top most correlated 50 genes generated in revised Fig.5f.

14. My last point relates to editing. There are numerous spelling and grammatical errors that need to be addressed throughout the manuscript and figures.

We apologize for the spelling and grammatical errors. We have carefully proofread the revised manuscript.

Reviewer #2 (Remarks to the Author):

Comments to the author:

Zheng et al. are the first to comprehensively analyze the tumor microenvironment of esophageal squamous cell carcinoma (ESCC) based on single-cell RNA-seq and coupled T cell receptor (TCR)-seq of 80,787 immune cells derived from seven surgically removed ESCC tumors and their matched adjacent normal tissues. The authors observe a correlation between immunosuppression and the failure of immunosurveillance in ESCC, and eventually focus on the interaction of macrophages and NK cells which may contribute to the immunosuppressive status. Moreover, they identified a six-gene signature that is associated with poor survival in ESCC and other cancers. Overall, the group shows findings that would be of interest to scientists in the field of ESCC due to increasing interests to profile the immune status of ESCC in order to evaluate the application of current checkpoint blockade therapies and to develop innovative immunotherapies.

We appreciate the reviewer for highlighting major findings of our study and concluding that our results would be of interest to scientists in the field.

However, with regard to the broader community and the wider scope of single-cell research in immunotherapy, the work does not influence thinking in the field. Most of the analyses are descriptive and sometimes lack a clear rationale.

Major comments:

1. Overall, the paper lacks a bit novelty and the data have not been interpreted sufficiently. The authors are the first to profile the tumor microenvironment in ESCC, but very similar approaches and analyses as seen across other cancer types. Some conclusions are not well underpinned, unclear or sometimes missing, e.g.:

- Immunosuppression has already been indicated to be responsible for failure of immunosurveillance before (1)*
- The proposed six-gene signature which they found to be associated with poor survival in ESCC contains genes which have already been shown to be predictive in ESCC and other cancer types, such as IL-10 and CD274 (PD-L1) (2,3)*
- The techniques mentioned are novel in ESCC, however they could have used additional algorithms to further dig into the single cell data, e.g. to study cell interactions (NicheNet, CellPhoneDB...), copy number variation in cancer cells (InferCNV)...*

We thank the reviewer for pointing out some weaknesses in our original manuscript. We agree with the reviewer that our analyses in original manuscript were largely descriptive. It was our intent to provide an in depth look at the immune landscape of ESCC, one of the most common cancer types, because it has not been reported previously. Thus, the information we are providing here will be valuable for investigators interested in ESCC, as well as for the growing community of cancer immunology. For example, the reviewers suggested us to compare our data from ESCC to other cancers. Such a comparison fully depends on studies similar to ours and published previously in other tumor types. Our study will provide a similar opportunity for following cancer immunological studies, in addition to serving as a resource for immune-related studies in ESCC. Additionally, in this revised manuscript, as suggested by the reviewers, results from a large amount of new experiments, e.g. flow cytometry, multi-color IHC staining, *in vitro* co-culture, antibodies blocking experiments, and new analyses, e.g. WGCNA, InferCNV, Slingshot, and scTHI, were included. These new results not only verified our original findings but also provided additional information about mechanisms and functional consequences. We believe these new data have significantly improved our manuscript. We appreciate the reviewers for the help.

We agree with the reviewer that immunosuppression has already been indicated to be responsible for failure of immunosurveillance in a wide range of cancer types. However, cancers from different primary sites and with different pathologies all have their unique characteristics, as shown in Revised Fig. 1j. This is why a lot of research effort has been made to profile the immune landscape of every single major type of cancer. ESCC is among the most common cancer types, but tumor infiltrating immune cells in ESCC has not been systematically studied. Thus, our study is valuable for scientists in the field, as all reviewers agreed.

As the reviewer pointed out, our six-gene signature in original manuscript contains genes that have already been shown to be predictive in ESCC and other cancer types. To identify novel prognosis-associated gene sets, we developed a composite signature of 50 genes that are associated with myeloid leukocyte activation and activation of immune response, based on our scRNA-seq data. This signature is significantly associated with better prognosis in ESCC and in other tumor types with similar pathology, such as cervical squamous cell carcinoma and lung squamous cell carcinoma. Please find more details in response to Reviewer 1 Comment 13.

We appreciate the reviewer's suggestion of including additional algorithms to further dig into the single cell data. In the revised manuscript, we used additional algorithms, e.g. Slingshot to study the cell trajectories, scTHI to study cell

interactions, WGCNA to study the gene correlation and found a module gene that is correlated with ESCC survival. We also used SCENIC analysis to find the transcription factors that are important to regulation of monocyte/macrophage differentiation, and we used InferCNV for calling copy number variation in cancer cells.

Revised Fig. 1j: Comparison of the fractions of T cells, B cells, NK, monocyte/macrophages in tumors from patients with ESCC, BRCA, LC, SKCM, HCC, COAD and KIRC. Skin Cutaneous Melanoma (SKCM); Breast invasive carcinoma (BRCA); Lung cancer (LC); Hepatocellular carcinoma (HCC); Colon cancer (COAD); Kidney cancer (KIRC).

2. It would be interesting if the authors also investigate malignant cells which they didn't cluster separately and annotated as "other". Gaining knowledge regarding the interactions between cancer cells and immune cells is fundamental to profile response to immune checkpoint therapy.

We thank for the reviewer the constructive suggestion. Following this suggestion, we analyzed the "other" cluster from our analysis, and found that a lot of cells had copy number variations, including amplifications or deletions those affecting known cancer genes. These cells were probably introduced by contamination during purification of CD45+ cell. It was included in supplementary Fig. 2. The estimated frequency of tumor cells in our analyzed cells is from 0.0274% to 0.487%, mainly in S150 and S149 samples. We understand the power of additional analyses based on "contaminated" tumor cells was limited due to the small number of tumor cells available. We are planning to conduct a follow up study to dissect the interaction between tumor cells and immune cells in ESCC. We included it in the discussion section. For the CNV analysis in the "other" cluster, please find more information in Review 1 Comment 3.

3. Their signatures for cytotoxicity, exhaustion, and regulatory T cells are based on the top 50 genes highly correlating with the expression of one single gene, i.e., *FGFBP2*, *LAG3*, and *FOXP3*, respectively (Fig. 2c). In other words, they conclude the functional state of cells by the expression of one anchor gene, which is not sufficient.

We thank the reviewer for this careful thought and agree that our approach of predicting cell state may directly and indirectly relies on one anchor gene. To avoid the bias introduced by this approach, we re-conducted analysis with gene signatures that have been published²³. As shown in Revised Fig. 2c and 2d, the results were consistent with the approach we used in original manuscript. The old approach still provides useful information and has been used in previous publications²⁴. Thus, we moved the old results to Revised supplementary Fig. 6d and 6e

Revised Fig. 2c and d: (c) Dot plot of representative cytotoxic, exhaustion, naïve, and Treg signatures in CD4 T cell clusters, Z-score normalized log₂ (count+1). (d) Dot plot of representative cytotoxic, exhaustion, and naïve signatures in CD8 T cell clusters, Z-score normalized log₂ (count+1).

4. Many of their clusters seem to be driven by cell cycle genes. Why didn't they regress for this? Same accounts for IFN- and stress-related markers.

We thank for the reviewer for looking deep into the data. These gene modules were not regressed in our original analysis in order to preserve most variances driven by biological processes and achieve better resolution when identifying subpopulations, which have been reported²⁴, like the proliferating clusters in CD4, CD8 and NK cells, respectively

(Revised Fig. 3k, l and m). In this revised manuscript, we performed analyses with regression of cell cycle-related, IFN-related, and stress-related genes, respectively. The results showed from the regression of these gene modules did not affect the percentage of T, B, NK, monocyte/macrophage, DC or mast cells in the tumor cells (Response Fig. 3). After such consideration, we decided not to perform such regression in the final analyses.

Response Fig. 3: Pie charts of cell-type fractions for ESCC tumor and normal tissues' infiltrating immune cells, colored by cell type, after regression cell cycle genes, IFN genes and stress genes, respectively.

5. The findings related to trajectory inference are highly dependent on the specific tool used to perform the analysis. It is strongly suggested to verify their findings for trajectory inference of T cells and monocytes/macrophages (Fig. 2, 5) using additional publicly available tools such as SCORPIUS, Slingshot, etc.

As the reviewer's suggested, we applied Slingshot (29914354) to the same dataset of CD8, Th and monocyte/macrophages trajectory analysis, respectively and obtained comparable results (Revised supplementary Fig. 6f, 6g, and 9b). CD8 T cells presented a continuous trajectory from the cytotoxic to exhaustion state. In Th cells, naïve cluster CD4-C1-CCR7 positioned at the opposite end of exhaustion clusters CD4-C5-STMN1, while CD4-C2-TCF7 and CD4-C3-CD40LG were located in-between, indicating their intermediate functional states. The monocyte/macrophages trajectory analysis showed the directionality from monocytes to M1 or M2 status.

Revised Supplementary Fig. 6f and g: The trajectory of Th cells (f) and CD8 T cells (g) state transition in a two-dimensional state-space inferred by Slingshot. Each dot corresponds to one single-cell, colored according to its cluster label. Arrows show the increasing directions of certain T cell properties.

Revised Supplementary Fig. 9b: The trajectory of Macro-C3-CSF1, Mono-C1-VCAN, Macro-C2-IL1RN, Macro-C1-IL6 state transition in a two-dimensional state-space inferred by Slingshot. Each dot corresponds to one single-cell, colored according to its cluster label. Arrows show the increasing directions of certain cell properties.

6. The authors point out the difference between a “pre-exhaustion” and “exhaustion” state. This resembles the concept of “progenitor” vs. “terminally exhausted” T cells. It is recommended to check the markers that are already published for

these specific states before concluding too quickly that the cells are “pre-exhausted”, namely solely based on a more moderate expression of immune checkpoints (p. 18).

We identified three clusters (CD8-C5-CCL5, CD8-C6-STMN1, and CD8-C7-TIGIT) of CD8 T cells expressing different levels of exhaustion markers, including PD-1, TIGIT, CTLA-4, TIM-3, and LAG-3. We suggested CD8-C5-CCL5 and CD8-C6-STMN1 are at pre-exhaustion states. We agree with the reviewer that this statement need to be backed by established “pre-exhausted” and “terminal-exhausted” markers. Distinguish stages of CD8 T cell exhaustion have been reported by studies in other cancer types, for examples, PD-1-inter, Eomes-low are markers of early exhaustion, and PD-1-hi, Eomes-hi are markers of terminal exhaustion⁹. We checked the expression of PD1 (PDCD1) and Eomes expression in these clusters. We found that CD8-C5-CCL5 expressed a lower level of PDCD1 and EOMES compared with CD8-C7-TIGIT, and CD8-C6-STMN1 was in the middle. It suggested that CD8-C5-CCL5 cells were at an early stage of exhaustion, while CD8-C7-TIGIT was terminal exhausted. CD8-C6-STMN1 was likely in a transition stage between CD8-C5-CCL5 and CD8-C7-TIGIT. The exhaustion at reversible stage can be rescued by checkpoint blockade and but not at permanent stage that are established by epigenetic alterations^{10,11}. We checked the expression levels of TOX and NFATC2 in CD8 T cell clusters. Both genes participate in establishing epigenetic programs to install permanent exhaustion status. The revised Fig. 2i showed that cluster 7 expressed much level of TOX, NFATC2, suggesting terminal exhaustion state.

Revised Fig. 2i: Violin plot showed the *PDCD1*, *NFATC2*, *EOMES* and *TOX* in CD8-C5-CCL5, CD8-C6-STMN1, and CD8-C7-TIGIT clusters. The expression was measured as the log₂ (count+1).

7. When conducting pathway enrichment analysis of *Lamp3*⁺ dendritic cells, they found these cells to be enriched in several pathways (pathways of cytokine-mediated signaling transduction, DC cell differentiation, leukocyte activation, membrane trafficking, and antigen processing and presentation). However, they conclude that these cells are tolerogenic without giving the substantial proof needed for this conclusion (p.14 + fig. 5i and Extended 5f).

The *Lamp3*⁺ dendritic cells are likely multi-functional^{4,25}. As the reviewer suggested, we checked the tolerogenic markers, like *IDO1*, *EBI3*, *CD274*, *IL10*, and found they were highly expressed in this cluster compared to other DC clusters (Revised Fig. 6d). We further analyzed the *LAMP3*⁺DC in additional samples by flow cytometry and multi-color IHC staining. Data showed that *LAMP3*⁺DC expressed much higher *CD83*, *CCR7* and *PDL1* than *LAMP3*⁻DC (Revised Fig. 6e and f), suggesting the maturation, migration and regulation ability of *LAMP3*⁺DC. Multi-color IHC staining also validated the existence of tolerogenic DC that express *CD11C*, *LAMP3*, *PDL1*, and *IDO* (Revised Fig. 6g). *IFN* γ and LPS could stimulate DC *PDL1* and *IDO1* expression (Revised Fig. 6h), and after *IFN* γ and LPS stimulation, DC had much more ability to induce *Foxp3* expression from the *CD4*⁺*CD45RA*⁺ naïve T cells than medium (Revised Fig. 6i), and these data suggested that *IFN* γ and LPS may induce the tolerogenic DC *in vitro*.

Revised Fig. 6e-6i: (e) FACS analysis of LAMP3 and PDL1 expression in DC cells from adjacent and ESCC tumor tissues (n=5), ** $P < 0.01$, two-sided t-test. (f) CCR7 and CD83 expression in LAMP3+ and LAMP3- DC cells from adjacent and ESCC tumor tissues by FACS (n=4), error bars representing \pm SEM, * $P < 0.05$, two-sided t-test. (g) Detection of CD11C⁺LAMP3⁺PDL1⁺IDO⁺ DC in ESCC tissue by multi-color of IHC staining. (h) DC cells were stimulated with LPS and IFN γ or not for 24 hours, and PDL1, LAMP3 and IDO expression was analyzed, one of the three experiments was represent, error bars representing \pm SEM, * $P < 0.05$, two-sided t-test. (i) DC cells were stimulated with LPS and IFN γ or not for 24h and cultured with CD4⁺CD45RA⁺ naïve T cells for 4 days, FOXP3 expression was measured. One of the three experiments was represent, error bars representing \pm SEM, * $P < 0.05$, two-sided t-test.

8. *It is recommended to validate the interaction between myeloid, regulatory T cells, and NK cells by the use of other cell-interaction algorithms (such as NicheNet and CellPhoneDB) and immunohistochemistry if possible. Further validation is also required for the finding that macrophages have the potential to upregulate the function of regulatory T cells.*

We thank the reviewer for this constructive suggestion. Given this suggestion, we applied another methodology scTHI (<https://github.com/miccec/scTHI>) to predict cell-cell interaction, and obtained comparable results (Supplementary table 8).

Two interactions were selected for further validation studies: (1) IL-1 β in the macrophage and its receptor IL1R2 in Treg cells and (2) MHC in Tregs and LILRB1 in the macrophage. We have conducted flow cytometry, multi-color IHC staining, *in vitro* co-culture and antibody blocking experiments to validate the communication between macrophage and Treg through these two of ligand/receptor pairs and assess the functional consequences. Please find more details in response to Reviewer 1 Comment 12. These new data are presented in a re-organized Figure 7 in the revised manuscript.

9. *The statistics are poorly described and sometimes missing, e.g.:*

- *Inter-patient variation in biologic signatures: is the difference between S135 & S158 vs. S33 & S134 significant?*

(Extended Fig. 1c-f)

We apologize for not clearly explains these results. These results were used to illustrate the heterogeneity among samples. If comparing S135&S158 Vs S133&S134, the differences were statistically different. Based on the rank sum test, the P

value was $<10^{-30}$, 10^{-30} , and 10^{-15} , for hypoxia, inflammation response and TNFA-via NFkB pathways, respectively.

- Comparing % of each cell type between tumor and adjacent tissue: are these differences significant? (Fig. 1e)

We apologize for not clearly stating the statistics in this result. We have updated this result in the revised supplementary Fig.3a. There was no significant difference between each cell type in tumor and adjacent tissue, though some cell types had an obvious trend, such as the increasing T cell and monocyte/macrophage proportion but decreasing NK proportion in tumor tissue. We realized that this cohort of ESCC was very heterogeneous and the sample size was small, so we conducted experimental analysis with 12 pairs additional samples by FACS and IHC staining. Those analyses provided comparable results with improved statistics (Revised Supplementary Fig. 4 and 5).

Revised supplementary Fig. 3a: Bar plot showed the each cell type between tumor and adjacent tissue, P values were calculated by paired student t test.

Revised Supplementary Fig. 4: IHC stained of CD3 (T cell), CD56 (NK), CD20 (B cell), CD68 (macrophage), MPO (neutrophil) in adjacent and tumor tissue of ESCC (n=12), and analysis of positive of cell number in each vision. * $P < 0.05$; two-sided t-test.

Supplementary Fig. 5: FACS analysis of adjacent and ESCC tumor tissues (n=12) of immune subsets, including T cells, B cells, plasma, NK, DC, mast, neutrophils, and macrophages, the gating strategies (a) and the percentage (b) was shown.

* $P < 0.05$, ** $P < 0.01$; two-sided t-test.

- Comparing clonality between CD4 and CD8 T cells (Fig. 4d)

We apologize for not clearly stating the statistics, when compare clonality between CD4 and CD8 T cells, the Ranksum test $P=0.005$. We have updated these descriptions in the revised figure legends.

10. Language is sometimes problematic in this manuscript and not consistent, e.g.:

- genes in small vs. capital letters, e.g. p. 14 *Lamp3+* vs. *LAMP3+*
- p. 6 *NFKB* pathways -> *NFKB* pathways
- p. 8 *suggested a* -> *suggested a*
- p.15 *while high* -> *while high*
- p. 16, 20: *prognosis biomarker* -> *prognostic biomarker*
- p. 18: *tothe* -> *to the*
- ...

We thank the reviewer for carefully reading our manuscript and apologize for inconsistencies and errors. The missing space was likely due to the using of different versions of Microsoft Word program during our manuscript drafting. We apologize for not correcting them during final proofreading before submission. These errors have been corrected in the revised manuscript.

Minor points:

1. The authors compare the percentage of each cell type between tumor and adjacent tissue, but it should be taken into account that this depends on the site of biopsy.

We agree with the reviewer that the location of biopsy may significantly affect the results. During tissue collection, we tried to keep consistency and avoid potential cross contamination between tumor and adjacent tissues. All adjacent tissues were at least 5 cm away from the closest tumor tissues. We now included the cartoon and one photos taken during tissue collection to indicate locations of tumor and adjacent tissues (Revised Fig. 1a and Response's Fig. 2).

Revised Fig. 1a: Schematic diagram of the experimental design and analysis.

Response Fig.2. The position of tumor (Red circle) and adjacent tissue (yellow circles) in one representative sample.

2. *Wrong reference to the figure showing the proliferating ability of NK and T cells: Fig. 4i-k -> Fig. 3*

We thank reviewer for helping us finding the error. It has been corrected in the revised manuscript.

3. *The studied cohort of seven patients is very heterogenous, consisting patients with ESCC of very diverse stages IIA-IIIB which is another variable that may influence the results.*

We agree with the reviewer that this cohort of ESCC is very heterogeneous. It is a common challenge for studies with limited sample availability, such as ESCC, and for studies that are extremely costly to scale-up, such as single cell sequencing. During revision, we analyzed an additional samples including stages IIA-IIIB (Revised supplementary table 1) by flow cytometry and staining, a similar level of heterogeneity was observed. Please find more detailed information in response to Reviewer 1 Comment 1. It indicates that ESCC in general are immune heterogeneous.

4. *Their written intentions do not always correspond to their actual data. Regarding the CD4 trajectory inference, for example, they state that they will not include regulatory T cells in their analysis, however, this is not consistent with the data shown (Fig 2h: they included cluster 13 which expresses a high level of regulatory T cell genes).*

We apologize for the error leading to this inconsistency. Though C13-CD4-STMRN1 expressed FOXP3, but was much lower than CD4-C6-FOXP3 (Fig. 1b). However, CD4-C5-STMN1 expressed a high level of CXCL13, which commonly highly express in exhausted T cells^{23,24}, and this cluster was highly proliferation, also the signature score showed that C13-CD4-STMRN1 was classified as exhausted T cells, though some genes were overlapped with Treg cells (Revised Fig.2c,

2d, supplementary Fig. 6e, 6h). This cluster was identified as exhausted Th cells, and it was included in the Th cell trajectory inference. We have corrected it in this revised manuscript.

5. The paragraph starting on page 14 includes insufficient description of Fig 5i, but describes the results from extended figure 5f.

We apologize for the error. We have corrected it in this revised manuscript.

Reviewer #3 (Remarks to the Author):

The manuscript from Zheng et al., described the immune landscape in esophageal squamous cell carcinoma (ESCC). The authors profiled stages 2A-3B tumour and peritumoral tissue (5 cm away from tumour) from six male and one female patients. The patients were aged between 54-81 and had not received prior treatment. 3000-9000 cells per sample were analysed from CD45+ cells that were isolated by FACS.

The main limitation of the study is its descriptive nature and with little validation beyond FACS analysis for cytokine production of T cells. Some of the data descriptions were not accompanied by tissue context or mechanistic understanding of the immune landscape.

1. The study will be significantly enhanced by in situ validation of the differential proportion of immune cells e.g. exhausted T cells, NK cells and macrophage profile in tumour vs peritumour.

We thank the reviewer for the suggestion and agree that validation is required to make solid conclusions. During revision, we have conducted a large amount of validation experiments and analyses. These validations include IHC staining, flow cytometry analysis, multi-color IHC staining and re-analyzing published bulk RNAseq data. Thus, we experimentally validated the vast majority of our findings for proportions of immune cells. Please find more information in the response to Reviewer 1 Comment 1. We also verified the exhausted T cells, NK cells and macrophage in tumor and adjacent tissue using flow cytometry, see revised Fig. 3f, 3i and supplementary Fig. 9e, also verified tolerogenic DCs by flow cytometry

and multi-color IHC staining, see revised Fig. 6e and 6g. These data suggest that the enrichment of multiple types of immune suppressive cells in human ESCC microenvironment.

Revised Fig. 3f and i: (f) Flow cytometry measured CD8 T cells PD1 expression in ESCC and adjacent tissues (n=7) $*P < 0.05$, Student's *t*-test. (i) Flow cytometry measured NK cells NKG2A expression in ESCC and adjacent tissues (n=7) $*P < 0.05$, Student's *t*-test.

Revised supplementary Fig. 9e: Flow cytometry measured CD68+ macrophage CD163 and CD206 expression in ESCC and adjacent tissues (n=6) $*P < 0.05$, Student's *t*-test.

Revised Fig. 6e: FACS analysis of LAMP3 and PDL1 expression in DC cells from adjacent and ESCC tumor tissues (n=5), * $P < 0.05$, student's t-test.

Revised Fig. 6g: Detection of CD11C⁺LAMP3⁺PDL1⁺IDO⁺ DC in ESCC tissue by multi-color of IHC staining.

2. The cell-cell interaction predictions will be stronger with validation to support the claims Mac-Treg interactions and others. This can be showing the interacting cells in situ and looking at pathways that are up or downregulated in the interacting pairs to support the R:L interactions.

We thank the reviewer for the suggestion. To further support the R:L interactions in our analysis, we did additional validation. First, we use another widely used methodology, scTHI, to predict the cell-cell interaction, and found the most results comparable. Second, we chose two interactions between Mac-Treg for further experimental validation. The validation includes flow cytometry, in situ multi-color IHC staining, *in vitro* co-culture functional experiments, and antibody blocking experiments. These data may not only validate the crosstalk between Mac-Treg, but also suggested their contribution to the immune suppressive in tumor microenvironment. In this revised version, we re-organized the figure by including these new data. Please find more details about the validation studies in response to Reviewer 1 Comment 12.

3. The numbering of cell states/clusters is very difficult to follow as they are not necessarily in chronological order – a simplified numbering of CD4_1 to CD4_n in chronological order for the broad immune categories may be easier.

We thank the reviewer for the suggestion. We named each cluster by “cell type- cluster number-marker gene”, like CD4-C6-FOXP3. We think including the marker gene will help readers to follow better.

4. The authors can leverage existing scRNA-seq datasets from cancer to annotate and compare myeloid and T cells with

other cancers. These can illustrate the similarities and distinctions across cancers which will be more powerful. Zilionis et al., Immunity 2019; Maier et al., Nature 2020 and Zhang et al., Cell 2019 provide good datasets to compare and contrast. Maier and Zhang also describe the regulatory DC subset which the authors observe in their study. Providing this context will add depth to the data presented in this study

We thank the reviewer for the suggestion. Several recent studies have utilized scRNA-seq to describe tumor infiltrates in melanoma, non-small cell lung cancer (NSCLC), breast cancer, liver cancer and colorectal cancer^{1,12,23,24,26}, all of these have explored the intrinsic transcriptional programs of T cell infiltrates. In line with previous observations^{2,24}, our data showed that the exhaustion T cell pool does not form a discrete cell population but is part of a wide differentiation spectrum, spanning from early exhaustion toward terminally exhausted T cells. More interestingly, a strong clonal sharing of pre-exhaustion and exhaustion CD8 cells was observed, but not with cytotoxic CD8 cells. These observations suggested that transcriptional gradients contribute to T cell heterogeneity. Also, exhausted T cells were the major intratumoral proliferating immune cell compartment²⁴. Nevertheless, the exhaustion signature was dominant at tumor sites, but not in adjacent tissue, suggesting a locally induced differentiation process. Our study also confirmed results from previous studies²⁴ that Treg cells shared many genes with exhausted T cells, including regulatory molecules and many co-inhibitory and co-stimulatory receptors (e.g., TNFRSF9, CSF1, IL1R2, and TIGIT). However, Treg cells expressed much higher levels of FOXP3, IL2RA, and CTLA4 than exhausted T cells, while exhausted T cells expressed higher levels of TNF, HIF1A, PDCD1, IFN, CXCL13 and CD38. Recently, Oh, *et al* reported that human bladder tumor contain multiple clonally expanded cytotoxic CD4+T cells and mediate anti-tumor cytotoxicity⁷, in our study, we found the cytotoxic CD4 T cells (CD4-C4-IFIT3) existed and clonally expansion (Revised Fig. 2c and Fig. 4d). Furthermore, regulatory CD4+T cells were enriched and clonally expanded in bladder tumors like ours. These studies suggest the enrichment of exhaustion and Treg programs are universal features in many kinds of tumors.

Recently, some single cell RNA-seq papers had reported the myloid cells in the tumor microenvironment^{4,25,27}. We found that most of them consistent with our data. Zilionise, et al reported that human DC contain four distinct subsets, including hDC1, hDC2, hDC3 and pDC, hDC1, hDC2, and hDC3 were consistent with our cDC1 (DC-C1-CLEC9A), cDC2 (DC-C2-CLEC10A), Lamp3+DC (DC-C3-LAMP3) and pDC (DC-C5-CLEC4C), respectively. However, the monocyte derived DC (DC-C4-FCER1A) was absent in that paper. These differences may reflect a variation in DC states between different tumor tissues and/or the setting of analysis. Interestingly, we found that LAMP3+DC has multiple functions,

such as activation activity, migration activity and tolerogenic ability, and this subset were also found and reported by Maier and Zhang of the lung and liver cancers, suggesting the conserved myeloid cells exist in many tumors.

We agree with the reviewer that these comparisons provided additional depth to our data and thank for this constructive suggestion.

5. M1/M2 are in vitro culture states, comparison with macrophages from other cancer data is much more appropriate.

Thanks for the reviewer's suggestion. Usually, macrophage activation has been classified into either a pro-inflammatory M1 state or an M2 state associated with the resolution of inflammation²⁸. In our dataset, using the published gene signatures of monocyte, M1, M2 and MDSC, macrophage clusters were identified, namely Macro-C1-IL6, representing M1 signature, Macro-C3-CSF1, representing M2 signature, MDSC-C1-C1QC and MDSC-C2-APOE representing MDSC signature^{29,30}, Macro-C2-IL1RN may represent the intermediate state, both enriched monocyte, M1 and M2 signature, Macro-C4-LILRB2, preferential and enrichment in adjacent tissue versus tumors was denoted as tissue resident macrophages (TRM). Zhang, et al have reported six macrophage cluster identified in hepatocellular carcinoma, M4-C1-THBS1 were enriched for signatures of MDSC, like ours MDSC-C1-C1QC and MDSC-C2-APOE; they also found the M4-C2-C1QA co-existence of M1 and M2 signatures¹. Lambrechts, *et al* have reported that macrophages in lung tumor microenvironment show rheostatic phenotypes and become M2 polarized³. Indeed, we found the M2 cluster Macro-C3-CSF1 was enriched in ESCC tissue than adjacent. We agree with the reviewer that M1/M2 does not necessarily reflect the status of tumor infiltrating macrophage. We observed a significant correlation between M1 and M2 signatures in macrophages (Revised supplementary Fig. 9c), the co-existence of M1 and M2 signatures indicated that TAMs were more complex than the classical M1/M2 model. We have included these discussions in revised manuscript (Page 22).

6. Trajectory inference can be misleading as immune cells could have easily been recruited from blood or altered state of resident cells and may not necessarily follow the linear order. For example, tumour associate macrophages may not arise from monocytes alone but be contributed by tissue resident macrophages and this cannot be ascertained in the type of pseudotime analysis. The claims from trajectory inferences need to be tempered or functionally validated as in the linear order.

We thank the reviewer for reminding us of the limitation of the trajectory inference. Though we distinguished the monocyte/macrophage clusters and annotated them by the published signatures, t-SNE plots and signature scores showed a poor separation of clusters, suggesting that they represent diverse cell states on graded scale rather than separate entities,

that are in line with the spectrum model of macrophage activity³¹. We picked four relatively typical clusters that represents the monocyte, M1, M2, intermediate state and did the trajectory analysis. It showed the increasing directions of certain cell properties that may represent the cell state transitions. As the reviewer suggested, immune cells could have easily been recruited from blood or altered state of resident cells and may not necessarily follow the linear order. Indeed, tumor associate macrophages may not arise from monocytes alone but be contributed by tissue resident macrophages. Recently, Zhang, et al reported that C1QC+ and SPP1+TAMs develop from unique tumor-infiltrating monocyte-like precursors, whereas SPP1+ TAMs might also derive from resident tissue macrophage in colon cancer¹. Thanks for the reviewer's suggestion; we have changed the claims of trajectory inferences much temper in the revised manuscript.

7. Peritumoral tissue (5 cm from tumour) should not be described as normal but as adjacent or peritumoral tissue due to cancer field effect.

We thank the reviewer for the suggestion and apologize for mixed usage of normal and adjacent tissues in the original manuscript. We have changed all normal tissue to adjacent tissue in the revised manuscript.

8. What doublet removal step was used in this study – given that the authors used a droplet platform this will be important to assess.

We applied Scrublet³² to remove putative hybrid transcriptomes occurring when two or more cells enter the same microfluidic droplet and receive the same barcode. Scrublet assigns each measured transcriptome a 'doublet score', which indicates the likelihood of being a hybrid transcriptome. We used a cluster-level approach to remove doublet clusters containing large number of potential doublet cells. Specifically, we removed the CD14+CD3+ cluster with a large fraction of potential monocyte-T cell doublets, which expressed both monocyte signature genes (CD14, S100A8, S100A9, S100A12) and T signature genes (CD3g, IL32, NKG7). We have included this important information in the revised manuscript methods.

9. Data availability code was not provided

The gene expression files for scRNA-seq are available at the Gene Expression Omnibus database with the access number GSE145370. Example of R Scripts for data process is available through <http://github.com/xinhua-lab/sc-hESCC>. Detailed

information will be available from the corresponding authors upon reasonable request. We include this information in the revised manuscript method.

10. Details of what the scale relating to gene expression, scores are not stated on figure panels or not provided in legend for many panels e.g. 1c, 1d, 2b, 2c, 2f (what is expression level?), 5e, 5f, 6a, 6c, 6h, 6i

Sorry for missing this information. In the revised manuscript, they have been added in the figure legend.

11. What is 2e gated on? Is this excluding CD4+ T cells as the staining for CD8 looks very discrete.

As reviewer 1 suggested in his/her comment 7, we agree that this experiment is not closely related to this study and is far from being able to draw a solid conclusion. We have removed this result in the revised manuscript and thank you for the suggestion.

12. The authors should show the data in supplementary figure of adding anti-PD1 and anti-Lag3 to cultured primary cells on CD8 cytokine production or remove the statement altogether from the MS if stating data not shown.

We accepted the reviewer's suggestion and removed this section from the manuscript, since it has little connection with the major purpose of this study.

13. What is the relevance of proliferation data in 3i,j,k

We apologize for not clearly interpreting results. Here we computed a proliferation score of cell cycle genes that were previously shown to denote G1/S or G2/M phases^{33,34} and used it to infer the proliferation status of T and NK cell clusters. Interestingly, we found that CD4-C5-STMN1 in CD4, CD8-C6-STMN1 in CD8, and NK-C2-STMN1 in NK were enrichment of exhaustion genes, and all of them were highly proliferative (Revised Fig. 4k, l, and m), which was consistent with the recent research suggesting that dysfunctional T cells were the major intratumoral proliferating immune cell compartment²⁴. Thus, the proliferation status of each cluster of the immune cells is a part of immune landscape of ESCC we aimed to provide in this study.

14. The T cell analysis shows very little sharing between tumour and adjacent tissue. This doesn't seem to be clearly stated

We apologize for not clearly interpreting results. Indeed, we found sharing of TCR sequences among all clusters in CD4 cells, and all clusters within CD8 cells, with the exception of C2 (Fig. 4g and h). However, clonal T cells in cytotoxic, exhaustion and Treg cells shared limited TCR between tumor and adjacent tissues, especially the Treg cells (Supplementary Fig. 8d). These data suggested that T cells may recruit from peripheral and stimulated by local environment and expanded. We have clearly stated this in the manuscript page14

Revised Fig. 4g and h: Share TCR clone types between different clusters in CD4 T cells (g) and CD8 T cells (h). The lines connected the dots means the clusters shared TCR clone types, bar plot showed the shared TCR clone type number, dot only suggested the unique clone types.

Revised supplementary Fig. 8d: The distribution of CD8-C1-NKG7, CD8-C7-TIGIT, CD4-C6-FOXP3 cell clonotypes between tumor and adjacent tissues.

References

1. Zhang, L., *et al.* Single-Cell Analyses Inform Mechanisms of Myeloid-Targeted Therapies in Colon Cancer. *Cell* **181**, 442-459 e429 (2020).
2. Azizi, E., *et al.* Single-Cell Map of Diverse Immune Phenotypes in the Breast Tumor Microenvironment. *Cell* **174**, 1293-1308 e1236 (2018).
3. Lambrechts, D., *et al.* Phenotype molding of stromal cells in the lung tumor microenvironment. *Nat Med* **24**, 1277-1289 (2018).
4. Zhang, Q., *et al.* Landscape and Dynamics of Single Immune Cells in Hepatocellular Carcinoma. *Cell* **179**, 829-845 e820 (2019).
5. Zack, T.I., *et al.* Pan-cancer patterns of somatic copy number alteration. *Nat Genet* **45**, 1134-1140 (2013).
6. Zhang, Y. & Ertl, H.C. Starved and Asphyxiated: How Can CD8(+) T Cells within a Tumor Microenvironment Prevent Tumor Progression. *Front Immunol* **7**, 32 (2016).
7. Oh, D.Y., *et al.* Intratumoral CD4(+) T Cells Mediate Anti-tumor Cytotoxicity in Human Bladder Cancer. *Cell* **181**, 1612-1625 e1613 (2020).
8. Simoni, Y., *et al.* Bystander CD8(+) T cells are abundant and phenotypically distinct in human tumour infiltrates. *Nature* **557**, 575-579 (2018).
9. McLane, L.M., Abdel-Hakeem, M.S. & Wherry, E.J. CD8 T Cell Exhaustion During Chronic Viral Infection and Cancer. *Annu Rev Immunol* **37**, 457-495 (2019).
10. Scott, A.C., *et al.* TOX is a critical regulator of tumour-specific T cell differentiation. *Nature* **571**, 270-274 (2019).
11. Ghoneim, H.E., *et al.* De Novo Epigenetic Programs Inhibit PD-1 Blockade-Mediated T Cell Rejuvenation. *Cell* **170**, 142-157 e119 (2017).
12. Zheng, C., *et al.* Landscape of Infiltrating T Cells in Liver Cancer Revealed by Single-Cell Sequencing. *Cell* **169**, 1342-1356 e1316 (2017).
13. Tirosh, I., *et al.* Dissecting the multicellular ecosystem of metastatic melanoma by single-cell RNA-seq. *Science* **352**, 189-196 (2016).
14. Street, K., *et al.* Slingshot: cell lineage and pseudotime inference for single-cell transcriptomics. *BMC Genomics* **19**, 477 (2018).
15. Stubbington, M.J.T., *et al.* T cell fate and clonality inference from single-cell transcriptomes. *Nat Methods* **13**, 329-332 (2016).
16. Kumar, M.P., *et al.* Analysis of Single-Cell RNA-Seq Identifies Cell-Cell Communication Associated with Tumor Characteristics. *Cell Rep* **25**, 1458-1468 e1454 (2018).
17. Cillo, A.R., *et al.* Immune Landscape of Viral- and Carcinogen-Driven Head and Neck Cancer. *Immunity* **52**, 183-199 e189 (2020).
18. Fernandez, D.M., *et al.* Single-cell immune landscape of human atherosclerotic plaques. *Nat Med* **25**, 1576-1588 (2019).
19. Ritvo, P.G., *et al.* Tfr cells lack IL-2Ralpha but express decoy IL-1R2 and IL-1Ra and suppress the IL-1-dependent activation of Tfh cells. *Sci Immunol* **2**(2017).
20. Guo, X., *et al.* Publisher Correction: Global characterization of T cells in non-small-cell lung cancer by single-cell sequencing. *Nat Med* **24**, 1628 (2018).
21. De Simone, M., *et al.* Transcriptional Landscape of Human Tissue Lymphocytes Unveils Uniqueness of Tumor-Infiltrating T Regulatory Cells. *Immunity* **45**, 1135-1147 (2016).
22. Barkal, A.A., *et al.* Engagement of MHC class I by the inhibitory receptor LILRB1 suppresses macrophages and is a target of cancer immunotherapy. *Nat Immunol* **19**, 76-84 (2018).
23. Guo, X., *et al.* Global characterization of T cells in non-small-cell lung cancer by single-cell sequencing. *Nat Med* **24**, 978-985 (2018).
24. Li, H., *et al.* Dysfunctional CD8 T Cells Form a Proliferative, Dynamically Regulated Compartment within Human Melanoma. *Cell* **176**, 775-789 e718 (2019).
25. Maier, B., *et al.* A conserved dendritic-cell regulatory program limits antitumour immunity. *Nature* **580**, 257-262 (2020).

26. Sade-Feldman, M., *et al.* Defining T Cell States Associated with Response to Checkpoint Immunotherapy in Melanoma. *Cell* **175**, 998-1013 e1020 (2018).
27. Zilionis, R., *et al.* Single-Cell Transcriptomics of Human and Mouse Lung Cancers Reveals Conserved Myeloid Populations across Individuals and Species. *Immunity* **50**, 1317-1334 e1310 (2019).
28. Mantovani, A., Sozzani, S., Locati, M., Allavena, P. & Sica, A. Macrophage polarization: tumor-associated macrophages as a paradigm for polarized M2 mononuclear phagocytes. *Trends Immunol* **23**, 549-555 (2002).
29. Condamine, T., *et al.* Lectin-type oxidized LDL receptor-1 distinguishes population of human polymorphonuclear myeloid-derived suppressor cells in cancer patients. *Sci Immunol* **1**(2016).
30. Xu, H., *et al.* Notch-RBP-J signaling regulates the transcription factor IRF8 to promote inflammatory macrophage polarization. *Nat Immunol* **13**, 642-650 (2012).
31. Xue, J., *et al.* Transcriptome-based network analysis reveals a spectrum model of human macrophage activation. *Immunity* **40**, 274-288 (2014).
32. Wolock, S.L., Lopez, R. & Klein, A.M. Scrublet: Computational Identification of Cell Doublets in Single-Cell Transcriptomic Data. *Cell Syst* **8**, 281-291 e289 (2019).
33. Whitfield, M.L., George, L.K., Grant, G.D. & Perou, C.M. Common markers of proliferation. *Nat Rev Cancer* **6**, 99-106 (2006).
34. Macosko, E.Z., *et al.* Highly Parallel Genome-wide Expression Profiling of Individual Cells Using Nanoliter Droplets. *Cell* **161**, 1202-1214 (2015).

Reviewers' Comments:

Reviewer #1:

Remarks to the Author:

In this revised version of the manuscript, the authors have stuck their brief of providing a detailed atlas and analysis of the landscape of immune cells in ESCC. They have performed a significant number of experiments to validate their findings with flow cytometry, immunostaining and bulk analysis of external datasets. In large part, these concur with their findings, and they were able to explain the deficiencies when the data was not concurrent, which are predominantly technical issues. The data concerning the macrophage/DC and identification of a new prognostic signature based on a specific trajectory point is far more interesting than their original signature. Network analysis has also been validated, for which the authors should be commended (it could have been just as easy to remove this section). There are a few figure legends that the authors can make more readable or palatable to the reader (such as figure 5e, where they have named each cluster but its color, which I find complicated as I do not have a Pantone colorboard by my side when I review papers), but these are minor issues.

The only issue I have with this revised version is their use of trajectory analyses, of any description, when used in an 'immunologically illogical manner'. Unlike the use for cancer cells, among immune cells, lineages are very well defined, and trajectory analysis (usually derived from stem cell-differentiation pathways) requires logical application. The analyses performed here is to put cancer-associated fibroblasts in the same trajectory analysis as cancer cells and declare that fibroblasts are at the end point of EMT, which is a ludicrous notion. Similarly, the trajectory analyses for CD8 and CD4 suffer the same issue. I feel that the data is flawed when two cell types that have no relationship to each other are placed in the same trajectory UNLESS you have a common ancestry (as seen in other papers, but here they did not identify a naive population). If the authors insist on showing trajectory analysis, they should limit these to subsets, for example putting CD8-C5-CCL5, CD8-C6-STMN1 and CD8-C7-TIGIT on a trajectory makes biological sense. Similar trajectory can be constructed for CD4 specific subpopulations as well.

Reviewer #2:

Remarks to the Author:

The manuscript significantly improved and the authors did an excellent job in answering to my questions. However, I still have a few smaller comments that would really need to be carefully assessed for the paper to be accepted in Nature Communications.

- There is still a problem with the nomenclature of the T-cells. For instance, the authors identify early to even terminally exhausted CD8+ T-cells, but it looks as if these populations are still quite cytotoxic and express high INFG. Are these populations really exhausted? Terminally exhausted T-cells are not supposed to be cytotoxic. The CD8_NKG7 population is characterized by high GZMB expression, is it possible that these cells represent the T-EMRA (effector-memory recently-activated) T-cell population? Where are the naïve CD8+ T-cells (is it possible they co-cluster within the naïve CD4+ T-cell cluster)? Perhaps add CD4 and CD8 expression to the heatmap in figure 2B. Overall, the authors should carefully compare their T-cell nomenclature to the one established in the literature (e.g., Zemin Zhang in Nature 2020).

- Page 4 – the claim that a high frequency of Tregs were derived from naïve CD4 T cells" is quite strong? It is only based on the fact that clonal T cells from the cytotoxic, exhaustion and regulatory clusters share limited TCRs between tumor and adjacent tissue. The authors omitted Tregs from the trajectory analysis, it is therefore difficult to come to such conclusion.

Reviewer #3:
None

Editor's remarks:

In particular we would expect you to remove the trajectory analysis and revise the nomenclature of your immune cells. You also need to ensure that the data is publicly available. At the same time we ask that you edit your manuscript to comply with our policies and formatting requirements and to maximize the accessibility and therefore the impact of your work.

*In this revised manuscript, we removed the trajectory analyses on CD4 and CD8 T cells, based on Reviewer 1's request. We revised the nomenclature of immune cells clusters and addressed all specific comments raised by Reviewer 2. We have also edited the manuscript to comply with the policies and formatting requirements of *Nature Communication*.*

Reviewer #1, expert in GI tract cancer (Remarks to the Author):

In this revised version of the manuscript, the authors have stuck their brief of providing a detailed atlas and analysis of the landscape of immune cells in ESCC. They have performed a significant number of experiments to validate their findings with flow cytometry, immunostaining and bulk analysis of external datasets. In large part, these concur with their findings, and they were able to explain the deficiencies when the data was not concurrent, which are predominantly technical issues. The data concerning the macrophage/DC and identification of a new prognostic signature based on a specific trajectory point is far more interesting than their original signature. Network analysis have also been validated, for which the authors should be commended (it could have been just as easy to remove this section. There a few figure legend that the authors can make more readable or palatable to the reader (such as figure 5e, where there have named each cluster but its color, which I find complicated as I do not have a Pantone colorboard by my side when I review papers), but these are minor issues.

We appreciate the reviewer for recognizing and highlighting the improvement of the revised manuscript.

The only issue I have with this revised version is their use of trajectory analyses, of any description, when used in an 'immunologically illogical manner'. Unlike the use for cancer cells, among immune cells, lineages are very well defined,

and trajectory analysis (usually derived from stem cell-differentiation pathways) requires logical application. The analyses performed here is to put cancer associated fibroblasts in the same trajectory analysis as cancers cells and declare that fibroblasts are at the end point of EMT, which is a ludicrous notion. Similarly, the trajectory analyses for CD8 and CD4 suffer the same issue. I feel that the data is flawed when two cells types that have no relationship to each other are placed in the same trajectory UNLESS you have a common ancestry (as seen in other papers, but here they did not identify a naive population). If the authors insist on showing trajectory analysis, they should limit these to subsets, for example putting CD8-C5-CCL5, CD8-C6-STMN1 and CD8-C7-TIGIT on a trajectory makes biological sense. Similar trajectory can be constructed for CD4 specific subpopulations as well.

We thank the reviewer and agree with the suggestion. The trajectory analyses on CD4 and CD8 cells are largely arbitrary. Therefore, we have removed them completely in this new version.

Reviewer #2, expert in single cell sequencing (Remarks to the Author):

The manuscript significantly improved and the authors did an excellent job in answering to my questions.

We appreciate the reviewer for recognizing the improvement of the manuscript and our efforts during revision.

However, I still have a few smaller comments that would really need to be carefully assessed for the paper to be accepted in Nature Communications.

- There is still a problem with the nomenclature of the T-cells. For instance, the authors identify early to even terminally exhausted CD8+ T-cells, but it looks as if these populations are still quite cytotoxic and express high INFG. Are these populations really exhausted ? Terminally exhausted T-cells are not supposed to be cytotoxic. The CD8_NKG7 population is characterized by high GZMB expression, is it possible that these cells represent the T-EMRA (effector-memory recently-activated) T-cell population ? Where are the naïve CD8+ T-cells (is it possible they co-cluster within the naïve CD4+ T-cell cluster) ? Perhaps add CD4 and CD8 expression to the heatmap in figure 2B. Overall, the

authors should carefully compare their T-cell nomenclature to the one established in the literature (e.g., Zemin Zhang in Nature 2020).

We thank for the reviewer's suggestions.

CD8-C7-TIGIT cells expressed the highest level of exhausted markers, like PDCD1 (PD1), HAVCR2 (TIM3), LAG3, TIGIT and CTLA4, suggesting they are exhausted CD8 T cells. However, they still expressed high level of IFNG and other cytotoxic genes. We agree with the reviewer that terminally exhausted T cells are not supposed to be cytotoxic. Thus CD8-C7-TIGIT cells are hard to be defined as terminally exhausted T cells. We have changed the descriptions in the revised manuscript; CD8-C7-TIGIT cells are described as exhausted CD8 T cells.

The CD8-C1-NKG7 cell expressed high level of GZMB and NKG7, low expression of CCR7 and SELL, and as the reviewer suggested, it may represents the T_{EMRA} cells, based on the T_{EMRA} markers (Fig 2B), we changed the descriptions in the revised manuscript. We thank the reviewer for the suggestion.

We did not detect a significant population of naïve CD8 T cells. To exclude the possibility that they were co-clustered with naïve CD4 T cells like reviewer suggested, we checked the expression of CD4 and CD8a in the CD4-C1-CCR7 naïve CD4 T cells cluster. We found no CD8a expression in the cells in this cluster, suggesting the co-clustering with naïve CD4 T cells was not the reason of missing naïve CD8 T cells. This data was included in the Fig 2B in the revised manuscript. It is possible that naïve CD8 T cells less frequently infiltrate into esophagus or they are quickly activated by the local environment, making them less likely to be detected as an independent population. Additional studies are needed to validate this phenomenon and identify the mechanism.

- Page 4 – the claim that a high frequency of Tregs were derived from naïve CD4 T cells” is quite strong? It is only based on the fact that clonal T cells from the cytotoxic, exhaustion and regulatory clusters share limited TCRs between tumor and adjacent tissue. The authors omitted Tregs from the trajectory analysis, it is therefore difficult to come to such conclusion.

We agree with the reviewer that the claim of Tregs derived from naïve CD4 T cells was solely based on sharing TCRs, thus it is not a solid conclusion. It only suggests potential common origins of these cells. We modified the manuscript accordingly.